# *SynCo*: Synthetic Hard Negatives for Contrastive Visual Representation Learning

## Abstract

Contrastive learning has become a dominant approach in self-supervised visual representation learning, but efficiently leveraging hard negatives, which are samples closely resembling the anchor, remains challenging. We introduce SynCo (***Sy***nthetic ***n***egatives in ***Co***ntrastive learning), a novel approach that improves model performance by generating synthetic hard negatives on the representation space. Building on the MoCo framework, SynCo introduces six strategies for creating diverse synthetic hard negatives "*on-the-fly*" with minimal computational overhead. SynCo achieves faster training and strong representation learning, surpassing MoCo-v2 by **+0.4%** and MoCHI by **+1.0%** on ImageNet ILSVRC-2012 linear evaluation. It also transfers more effectively to detection tasks achieving strong results on PASCAL VOC detection (57.2% AP) and significantly improving over MoCo-v2 on COCO detection (**+1.0%** $AP^{bb}$) and instance segmentation (**+0.8%** $AP^{msk}$). Our synthetic hard negative generation approach significantly enhances visual representations learned through self-supervised contrastive learning. Code will be made publicly available.

## 1 Introduction

Contrastive learning has emerged as a prominent approach in self-supervised learning, significantly advancing representation learning from unlabeled data. This technique, which discriminates between similar and dissimilar data pairs, has shown premise in visual representation tasks. Seminal works such as SimCLR (Chen et al., 2020b) and MoCo (He et al., 2020) established instance discrimination as a pretext task. These methods generate multiple views of the same data point through augmentation, training the model to minimize the distance between positive pairs (augmented views of the same instance) while maximizing it for negative pairs (views of different instances).

Despite its effectiveness, instance discrimination faces challenges. A key limitation is the need for numerous negative samples, often leading to increased computational costs. For example, SimCLR requires large batch sizes for sufficient negatives (Chen et al., 2020b). While approaches like MoCo address some issues through dynamic queues and momentum encoders (He et al., 2020; Chen et al., 2020c), they still face challenges in selecting and maintaining high-quality hard negatives. Some variations, like SimCo (Zhang et al., 2022a), take a different approach by removing both the momentum encoder and queue in favor of a dual temperature mechanism that modulates positive and negative sample distances differently in the InfoNCE loss.

Recent studies have highlighted the importance of carefully crafted data augmentations in learning robust representations (Chen et al., 2020b; Dwibedi et al., 2021; Tian et al., 2020b; Wang & Qi, 2022; Reed et al., 2021; Balestriero et al., 2023; Rojas-Gomez et al., 2024). These transformations likely provide more diverse, challenging copies

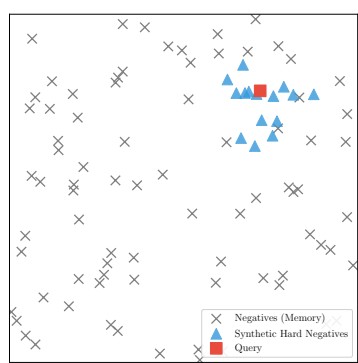

Figure 1: SynCo generates synthetic hard negatives for each query example like Kalantidis et al. (2020), *but better*.

of images, increasing the difficulty of the self-supervised task. This self-supervised task is a pretext problem (e.g., predicting image rotations (Gidaris et al., 2018) or solving jigsaw puzzles (Noroozi & Favaro, 2016)) designed to induce learning of generalizable features without explicit labels. Moreover, techniques that combine data at the pixel level (Zhang et al., 2017b; Yun et al., 2019) or at the feature level (Verma et al., 2018) have proven effective in helping models learn more resilient features, leading to improvements in both fully supervised and semi-supervised tasks.

The concept of challenging negative samples has been explored as a way to enhance contrastive learning models. These samples, which lie close to the decision boundary, are crucial for refining the model's discriminative abilities. Recent work like MoCHI (Kalantidis et al., 2020) has shown improvements by incorporating harder negatives. However, while the potential of hard negatives is clear, recent trends in AI have shifted focus toward large-scale foundation models (Bommasani et al., 2021; Awais et al., 2023), leaving this promising direction relatively unexplored. Yet, as Yann LeCun observed, "*if AI is a cake, self-supervised learning is the bulk of the cake*". We argue that revisiting and modernizing self-supervised approaches, particularly through innovative hard negative strategies, remains crucial for advancing AI systems.

In this paper, we present SynCo (***Sy**nthetic **n**egatives in **Co**ntrastive learning*), a novel approach to contrastive learning that leverages synthetic hard negatives to enhance the learning process. Building on the foundations of MoCo, SynCo introduces six distinct strategies for generating synthetic hard negatives, each designed to provide diverse and challenging contrasts to the model. These strategies include: interpolated negatives; extrapolated negatives; mixup negatives; noise-injected negatives; perturbed negatives; and adversarial negatives. By incorporating these synthetic samples, SynCo aims to push the boundaries of contrastive learning, improving both the efficiency and effectiveness of the training process.

A toy illustration of our synthetic hard negative generation approach is shown in Figure 1, which displays a t-SNE visualization of random embeddings projected onto the unit hypersphere. For any given positive query (red square), we notice that the memory bank (gray marks) predominantly stores easy negatives with relatively few challenging ones—meaning most negatives are positioned too far away to provide meaningful gradients for the contrastive loss. Our approach focuses on leveraging only the most challenging negatives (determined by their similarity scores with the query) to create new synthetic negatives that are both difficult and diverse (blue triangles).

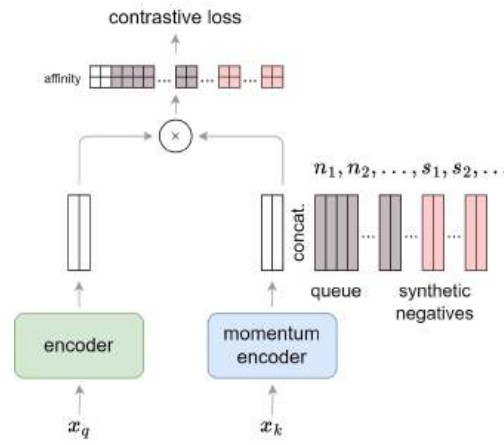

Figure 2: SynCo extends MoCo (He et al., 2020; Chen et al., 2020c) by introducing synthetic hard negatives generated "*on-the-fly*" from a memory queue. The process begins with two augmented views of an image, $\mathbf{x}_q$ and $\mathbf{x}_k$, processed by an encoder and a momentum encoder, respectively, producing feature vectors $\mathbf{q}$ and $\mathbf{k}$. The memory queue holds negative samples $\mathbf{n}_1, \mathbf{n}_2, \ldots$, which are concatenated with synthetic hard negatives $\mathbf{s}_1, \mathbf{s}_2, \ldots$ generated using the SynCo strategies. These combined negatives are used to compute the affinity matrix, which, together with the positive pair (query $\mathbf{q}$ and key $\mathbf{k}$), contributes to the InfoNCE loss calculation.

The main **contributions** of our work are as follows:

- We introduce SynCo, a contrastive learning framework that improves representation learning by leveraging synthetic hard negatives. SynCo enhances model discriminative capabilities by generating challenging negatives "*on-the-fly*" from a memory queue, using six distinct strategies targeting different aspects of the feature space. This process improves performance without significant computational increases, achieving faster training and stronger representation learning.

- We empirically show improved downstream performance on ImageNet ILSVRC-2012 by incorporating synthetic hard negatives, demonstrating improvements in both linear evaluation and semi-supervised learning tasks.

- We show that SynCo learns stronger representations by measuring their transfer learning capabilities COCO and PASCAL VOC detection, where it outperforms both the supervised baseline and MoCo.

The paper is structured as follows: Section 2 reviews related work; Section 3 explores hard negatives in contrastive learning; Section 4 introduces our synthetic hard negatives method; Section 5 presents experimental results; Section 6 offers discussion and analysis; and Section 7 concludes the paper.

## 2    Related Work

### 2.1    Contrastive Learning

Recent contrastive learning methods focus on instance discrimination as a pretext task, treating each image as its own class (Chen et al., 2020b; He et al., 2020). The core principle involves bringing an anchor and a "positive" sample closer in the representation space while pushing the anchor away from "negative" samples (Khosla et al., 2021). Positive pairs are typically created through multiple views of each data point (Tian et al., 2020b; Caron et al., 2020), using techniques such as color decomposition (Tian et al., 2020a), random augmentation (Chen et al., 2020b; He et al., 2020), image patches (van den Oord et al., 2019), or student-teacher model representations (Grill et al., 2020; Caron et al., 2021; Oquab et al., 2023). The common training objective, based on InfoNCE (van den Oord et al., 2019) or its variants (Chen et al., 2020b; Dwibedi et al., 2021; Tomasev et al., 2022; Yeh et al., 2022), aims to maximize mutual information (Hjelm et al., 2019; Bachman et al., 2019), necessitating numerous negative pairs. While some approaches like SimCLR use large batch sizes (Chen et al., 2020b) to address this, others like MoCo (He et al., 2020; Chen et al., 2020c), PIRL (Misra & van der Maaten, 2019), and InstDis (Wu et al., 2018) employ memory structures. Recent advancements explore strategies such as regularizers (Mitrovic et al., 2020; Bardes et al., 2022a; Zhu et al., 2022; Bardes et al., 2022b) or prevent model collapse via redundancy reduction (Zbontar et al., 2021; Bandara et al., 2023). Some methods like SimSiam and BYOL eliminate negative samples through asymmetric Siamese structures or normalization (Grill et al., 2020; Chen & He, 2020; Caron et al., 2021; Oquab et al., 2023). Approaches such as LA (Zhuang et al., 2019) and PCL (Li et al., 2021a) address the false-negative pair issue, while DCL (Yeh et al., 2022) further improves representation learning by separating the learning of features and metrics into two distinct phases.

### 2.2    Hard Negatives

Hard negatives are critical in contrastive learning as they improve the quality of visual representations by helping to define the representation space more effectively. These challenging yet relevant samples are harder to distinguish from the anchor point, enabling the model to better differentiate between similar features. The use of hard negatives involves selecting samples that are similar to positive samples but different enough to aid in learning distinctive features. Dynamic sampling of hard negatives during training prevents the model from easily minimizing the loss, enhancing its learning capabilities (He et al., 2020; Chen et al., 2020b). Various approaches have been proposed to leverage hard negatives effectively. For instance, MoCo (He et al., 2020) utilizes a dynamic queue and momentum-based encoder updates to maintain fresh and challenging negatives throughout training. Other methods, such as SimCLR (Chen et al., 2020b) and InfoMin (Tian et al., 2020b), suggest adjusting the difficulty of negative samples by varying data augmentation techniques. This progressive increase in task difficulty benefits the training process. Building on these ideas, MoCHI (Kalantidis et al., 2020) has explored integrating hard negative mixing into existing frameworks to further improve performance. By employing these methods, models become more adept at handling detailed and complex tasks, ensuring each negative sample significantly contributes to optimizing learning outcomes and boosting overall model effectiveness.

# 3 Preliminaries

In this section, we establish the theoretical foundations of contrastive learning and analyze the critical role of hard negatives in representation learning.

## 3.1 Contrastive Learning

Contrastive learning seeks to differentiate between similar and dissimilar data pairs, often treated as a dictionary look-up where representations are optimized to align positively paired data through contrastive loss in the representation space (He et al., 2020). Given an image $x$, and a distribution of image augmentation $\mathcal{T}$, we create two augmented views of the same image using the transformation $t_q, t_k \sim \mathcal{T}$, i.e., $x_q = t_q(x)$ and $x_k = t_k(x)$. Two encoders, $f_q$ and $f_k$, namely the query and key encoders, generate the vectors $\mathbf{q} = f_q(x_q)$ and $\mathbf{k} = f_k(x_k)$, respectively. The learning objective minimizes a contrastive loss using the InfoNCE criterion (van den Oord et al., 2019):

$$\mathcal{L}(\mathbf{q}, \mathbf{k}, \mathcal{Q}) = -\log \frac{\exp(\mathbf{q}^\top \cdot \mathbf{k}/\tau)}{\exp(\mathbf{q}^\top \cdot \mathbf{k}/\tau) + \sum\limits_{\mathbf{n} \in \mathcal{Q}} \exp(\mathbf{q}^\top \cdot \mathbf{n}/\tau)} \tag{1}$$

Here, $\mathbf{k}$ is $f_k$'s output from the same augmented image as $\mathbf{q}$, and $\mathcal{Q} = \{\mathbf{n}_1, \mathbf{n}_2, \ldots, \mathbf{n}_K\}$ includes outputs from different images, representing negative samples of size $K$. The temperature parameter $\tau$ adjusts scaling for the $\ell_2$-normalized vectors $\mathbf{q}$ and $\mathbf{k}$. The key encoder $f_k$ can be updated in two ways. In the synchronized update approach, $f_k$ is updated synchronously with $f_q$, maintaining identical weights throughout training (Chen et al., 2020b). Alternatively, a momentum update scheme can be employed, where $f_k$ is updated using the equation: $\theta_k \leftarrow m \cdot \theta_k + (1-m) \cdot \theta_q$ (He et al., 2020). Here, $\theta_k$ and $\theta_q$ are the parameters of $f_k$ and $f_q$ respectively, and $m \in [0,1]$ is the momentum coefficient. This momentum approach allows $f_k$ to evolve more slowly, providing more consistent negative samples over time and potentially stabilizing the learning process. The memory bank $\mathcal{Q}$ can be defined in various ways, such as an external memory of all dataset images (Misra & van der Maaten, 2019; Tian et al., 2020a; Wu et al., 2018), a queue of recent batches (He et al., 2020), or the current minibatch (Chen et al., 2020b). Recent analysis (Miles & Mikolajczyk, 2024) has shown that the projection head's normalization significantly influences training dynamics and representation quality

The gradient of the contrastive loss in Equation (1) with respect to the query $\mathbf{q}$ is given by:

$$\frac{\partial \mathcal{L}(\mathbf{q}, \mathbf{k}, \mathcal{Q})}{\partial \mathbf{q}} = -\frac{1}{\tau} \left( (1 - p_k) \cdot \mathbf{k} - \sum_{\mathbf{n} \in \mathcal{Q}} p_n \cdot \mathbf{n} \right) \quad \text{where} \quad p_{z_i} = \frac{\exp(\mathbf{q}^\top \cdot \mathbf{z_i}/\tau)}{\sum_{j \in Z} \exp(\mathbf{q}^\top \cdot \mathbf{z_j}/\tau)} \tag{2}$$

with $\mathbf{z_i}$ being a member of the set $\mathcal{Q} \cup \{\mathbf{k}\}$. The positive and negative logits contribute to the loss similarly to a $(K+1)$-way cross-entropy classification, with the key logit representing the query's latent class (Arora et al., 2019).

## 3.2 Understanding Hard Negatives

The effectiveness of contrastive learning approaches hinges critically on the utilization of hard negatives (Arora et al., 2019; Hadsell et al., 2006; Iscen et al., 2018; Mishchuk et al., 2017; Wu et al., 2018; Kalantidis et al., 2020). Current approaches face significant challenges in efficiently leveraging these hard negatives. Sampling from within the same batch necessitates larger batch sizes (Chen et al., 2020b; 2021). Conversely, maintaining a memory bank containing representations of the entire dataset incurs substantial computational overhead in keeping the memory up-to-date (Misra & van der Maaten, 2019; Wu et al., 2018; He et al., 2020; Chen et al., 2020c). These limitations underscore the need for more efficient strategies to generate and utilize hard negatives in contrastive learning frameworks.

**Hardness of negatives.** The "hardness" of negative samples, defined by their similarity to positive samples in the representation space, determines how challenging they are for the model to differentiate, directly

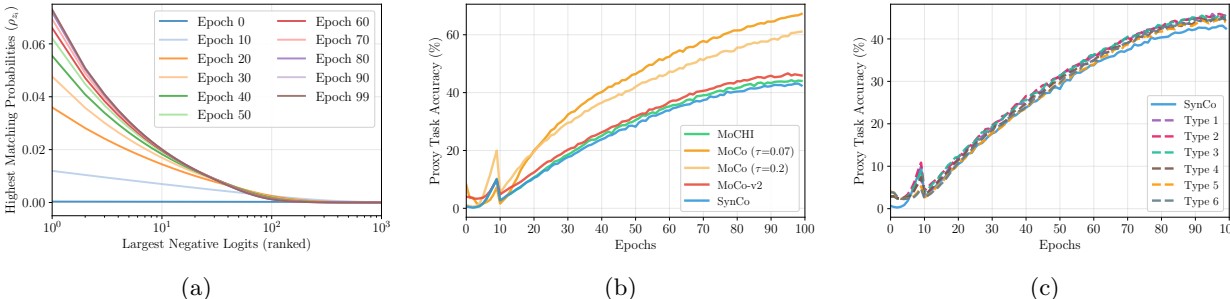

Figure 3: (a) Histogram of the top 1024 matching probabilities $p_{z_i}$, $z_i \in \mathcal{Q}$ for MoCo-v2 on ImageNet-100, over various epochs. Logits are organized in descending order, and each line indicates the mean matching probability across all queries Kalantidis et al. (2020). (b) Performance comparison of MoCo, MoCo-v2, MoCHI, and SynCo on ImageNet-100 in terms of accuracy on the proxy task (percentage of queries where the key is ranked higher than all negatives). (c) Performance comparison of SynCo under various configurations on ImageNet-100 in terms of accuracy on the proxy task.

impacting the effectiveness of the contrastive learning process. Figure 3a illustrates the evolution of negative sample hardness during MoCo-v2 training. Initially, the distribution of these probabilities is relatively uniform. However, as training progresses, a clear trend emerges: fewer negatives contribute significantly to the loss function. This observation suggests that the model rapidly learns to distinguish most negatives, leaving only a small subset that remains challenging. Such a phenomenon underscores the importance of maintaining a diverse pool of hard negatives throughout the training process to sustain effective learning (Kalantidis et al., 2020).

**Difficulty of the proxy task.** The difficulty of the proxy task in contrastive learning, typically defined by the self-supervised objective, significantly influences the quality of learned representations. Figure 3b compares the proxy task performance of MoCo and MoCo-v2 on ImageNet-100, measured by the percentage of queries where the key ranks above all negatives. Notably, MoCo-v2, which employs more aggressive augmentations, exhibits lower proxy task performance compared to MoCo, indicating a more challenging learning objective. Paradoxically, this increased difficulty correlates with improved performance on downstream tasks such as linear classification (Kalantidis et al., 2020). Additionally, Figure 3c demonstrates how SynCo's performance varies under different configurations, providing insights into the optimal parameter settings for balancing proxy task difficulty and representation quality. This counterintuitive relationship between proxy task difficulty and downstream performance suggests that more challenging self-supervised objectives can lead to the learning of more robust and transferable representations, motivating the development of strategies to dynamically modulate task difficulty during training.

## 4 Synthetic Hard Negatives in Contrastive Learning

In this section, we present an approach for generating synthetic hard negatives in the representation space using **six** distinct strategies. Building on MoCHI, we propose *four* additional strategies for generating synthetic hard negatives to explore complementary aspects of the representation space. A toy example of the proposed synthetic hard negative generation is presented in Figure 4. We refer to our proposed approach as SynCo ("***Sy***nthetic ***n***egatives in ***Co***ntrastive learning").

### 4.1 Generating Synthetic Hard Negatives

Let $\mathbf{q}$ represent the query image, $\mathbf{k}$ its corresponding key, and $\mathbf{n} \in \mathcal{Q}$ denote the negative features from a memory structure of size $K$. The loss associated with the query is computed using the logits $\ell(\mathbf{z_i}) = \mathbf{q}^\top \cdot \mathbf{z}_i / \tau$, which are processed through a softmax function. We define $\hat{\mathcal{Q}} = \{\mathbf{n}_1, \mathbf{n}_2, \ldots, \mathbf{n}_K\}$ as the ordered set of all negative features, where $\ell(\mathbf{n}_i) > \ell(\mathbf{n}_j)$ for all $i < j$, implying that the negative features are sorted based on

decreasing similarity to the query. The most challenging negatives are selected by truncating the ordered set $\hat{\mathcal{Q}}$, retaining only the first $N < K$ elements, denoted as $\hat{\mathcal{Q}}^N$.

**Interpolated synthetic negatives (type 1).** Building on MoCHI's (Kalantidis et al., 2020) foundation, our first strategy creates synthetic negatives through controlled interpolation between samples. This approach aims to generate features that lie in meaningful regions of the representation space between the query and existing hard negatives. For each query $\mathbf{q}$, we propose to generate $N_1$ synthetic hard negative features by mixing the query $\mathbf{q}$ with a randomly chosen feature from the $N$ hardest negatives in $\hat{\mathcal{Q}}^N$. Let $S^1 = \{\mathbf{s}_1^1, \mathbf{s}_2^1, \ldots, \mathbf{s}_{N_1}^1\}$ be the set of synthetic negatives to be generated. Then a synthetic negative feature $\mathbf{s}_k^1 \in S^1$ would be given by:

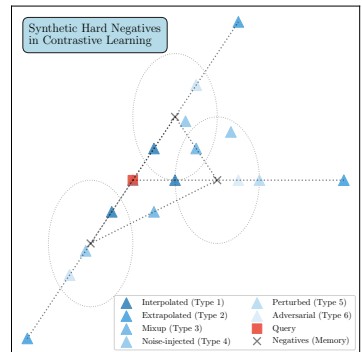

$$\mathbf{s}_k^1 = \alpha_k \cdot \mathbf{q} + (1 - \alpha_k) \cdot \mathbf{n}_i, \quad \alpha_k \in (0, \alpha_{\max}) \quad \text{where} \quad \mathbf{n}_i \in \hat{\mathcal{Q}}^N \quad (3)$$

and $\alpha_k$ is randomly sampled from a uniform distribution in the range $(0, \alpha_{\max})$. The resulting synthetic hard negatives are then normalized and added to the set of negative logits for the query. Interpolation creates a synthetic embedding that lies between the query and the negative in the representation space. We set $\alpha_{\max} = 0.5$ to guarantee

Figure 4: A toy example illustrating the six types of synthetic hard negatives generated by SynCo from a query point and three original negative points. More in Appendix F.

that the contribution of the query is always less than that of the negative. This is similar to the hardest negatives (type 2) of MoCHI (Kalantidis et al., 2020).

**Extrapolated synthetic negatives (type 2).** As a natural extension of interpolation, we propose extrapolation to explore the "opposite" direction in feature space. While this approach operates further from the decision boundary, we carefully control the exploration through coefficients to maintain an appropriate level of task difficulty. For each query $\mathbf{q}$, we propose to generate $N_2$ hard negative features by extrapolating beyond the query embedding in the direction of the hardest negative features. Similar to the interpolated method, we use a randomly chosen feature from the $N$ hardest negatives in $\hat{\mathcal{Q}}^N$. Let $S^2 = \{\mathbf{s}_1^2, \mathbf{s}_2^2, \ldots, \mathbf{s}_{N_2}^2\}$ be the set of synthetic negatives to be generated. Then a synthetic negative feature $\mathbf{s}_k^2 \in S^2$ would be given by:

$$\mathbf{s}_k^2 = \mathbf{n}_i + \beta_k \cdot (\mathbf{n}_i - \mathbf{q}), \quad \beta_k \in (1, \beta_{\max}) \quad \text{where} \quad \mathbf{n}_i \in \hat{\mathcal{Q}}^N \quad (4)$$

and $\beta_k$ is randomly sampled from a uniform distribution in the range $(1, \beta_{\max})$. These synthetic features are also normalized and used to enhance the negative logits. Extrapolation generates a synthetic embedding that lies beyond the query embedding in the direction of the hardest negative. We choose $\beta_{\max} = 1.5$.

**Mixup synthetic negatives (type 3).** Following MoCHI's (Kalantidis et al., 2020) effective strategy of mixing hard negatives, we incorporate their approach of combining pairs of challenging examples. For each query $\mathbf{q}$, we propose to generate $N_3$ hard negative features by combining pairs of the $N$ hardest existing negative features in $\hat{\mathcal{Q}}^N$. Let $S^3 = \{\mathbf{s}_1^3, \mathbf{s}_2^3, \ldots, \mathbf{s}_{N_3}^3\}$ be the set of synthetic negatives to be generated. Then a synthetic negative feature $\mathbf{s}_k^3 \in S^3$ would be given by:

$$\mathbf{s}_k^3 = \gamma_k \cdot \mathbf{n}_i + (1 - \gamma_k) \cdot \mathbf{n}_j, \quad \gamma_k \in (0, 1) \quad \text{where} \quad \mathbf{n}_i, \mathbf{n}_j \in \hat{\mathcal{Q}}^N \quad (5)$$

and $\gamma_k$ is randomly sampled from a uniform distribution in the range $(0, 1)$. The resulting synthetic hard negatives are then normalized and added to the set of negative logits for the query. Mixup combines pairs of the hardest existing negative features to create a synthetic embedding that represents a blend of challenging cases. This is similar to the hard negatives (type 1) of MoCHI (Kalantidis et al., 2020).

**Noise-injected synthetic negatives (type 4).** To prevent overfitting to specific negative patterns while maintaining the essential characteristics of hard negatives, we introduce controlled stochasticity through noise injection. For each query $\mathbf{q}$, we propose to generate $N_4$ hard negative features by adding Gaussian noise to the hardest negative features. Using the top $N$ hardest negatives $\hat{\mathcal{Q}}^N$, let $S^4 = \{\mathbf{s}_1^4, \mathbf{s}_2^4, \ldots, \mathbf{s}_{N_4}^4\}$ be the set of synthetic negatives to be generated. Then a synthetic negative feature $\mathbf{s}_k^4 \in S^4$ would be given by:

$$\mathbf{s}_k^4 = \mathbf{n}_i + \mathcal{N}(\mathbf{0}, \sigma^2 \cdot \mathbf{I}) \quad \text{where} \quad \mathbf{n}_i \in \hat{\mathcal{Q}}^N \tag{6}$$

and $\mathcal{N}(\mathbf{0}, \sigma^2 \cdot \mathbf{I})$ represents Gaussian noise with standard deviation $\sigma$ (where $\mathbf{I}$ is the identity matrix). The noisy negatives are normalized before being used in the loss calculation. Noise injection adds Gaussian noise to the hardest negative features, resulting in a synthetic embedding with added randomness.

**Perturbed synthetic negatives (type 5).** Drawing inspiration from adversarial training (Mehrabi et al., 2021), we introduce perturbed synthetic negatives that use gradient-based perturbations with variable magnitudes. For each query $\mathbf{q}$, we propose to generate $N_5$ hard negative features by perturbing the embeddings of the hardest negative features. Given the top $N$ hardest negatives $\hat{\mathcal{Q}}^N$, let $S^5 = \{\mathbf{s}_1^5, \mathbf{s}_2^5, \ldots, \mathbf{s}_{N_5}^5\}$ be the set of synthetic negatives to be generated. Then a synthetic negative feature $\mathbf{s}_k^5 \in S^5$ would be given by:

$$\mathbf{s}_k^5 = \mathbf{n}_i + \delta \cdot \nabla_{\mathbf{n}_i} \text{sim}(\mathbf{q}, \mathbf{n}_i) \quad \text{where} \quad \mathbf{n}_i \in \hat{\mathcal{Q}}^N \tag{7}$$

and $\text{sim}(\cdot, \cdot)$ is the similarity function and $\delta$ controls the perturbation magnitude. The perturbed embeddings are then normalized and added to the negative logits. Perturbation modifies the embeddings of the hardest negative features based on the gradient of the similarity function, creating synthetic negatives that are slightly adjusted to be more challenging for the model. This approach offers greater flexibility than fixed interpolation, as it generalizes to arbitrary similarity functions and can generate negatives of varying hardness.

**Adversarial synthetic negatives (type 6).** While similar in concept to type 5, adversarial synthetic negatives differ fundamentally in their gradient scaling approach. For each query $\mathbf{q}$, we propose to generate $N_6$ hard negative features by applying adversarial perturbations to the hardest negative features to maximize their similarity to the query embeddings. Using the top $N$ hardest negatives $\hat{\mathcal{Q}}^N$, let $S^6 = \{\mathbf{s}_1^6, \mathbf{s}_2^6, \ldots, \mathbf{s}_{N_6}^6\}$ be the set of synthetic negatives to be generated. Then a synthetic negative feature $\mathbf{s}_k^6 \in S^6$ would be given by:

$$\mathbf{s}_k^6 = \mathbf{n}_i + \eta \cdot \text{sign}(\nabla_{\mathbf{n}_i} \text{sim}(\mathbf{q}, \mathbf{n}_i)) \quad \text{where} \quad \mathbf{n}_i \in \hat{\mathcal{Q}}^N \tag{8}$$

and $\eta$ controls the perturbation magnitude. The perturbed embeddings are normalized and added to the negative logits. Adversarial hard negatives apply adversarial perturbations to the hardest negative features, specifically altering them to maximize their similarity to the query embeddings, thereby producing the most challenging contrasts. Where type 5 allows variable perturbation sizes, type 6 enforces unit magnitude through the sign function, creating consistently challenging contrasts.

## 4.2 Integrating Synthetic Hard Negatives into the Contrastive Loss

The synthetic hard negatives generated are integrated into the contrastive learning process by modifying the InfoNCE loss. Let $\mathcal{S} = \bigcup_{i=1}^{6} S^i$ represent the concatenation of all synthetic hard negatives, where $S^i$ is the set of synthetic negatives generated by the $i$-th strategy. This combined set of synthetic negatives augments the original negatives $\mathcal{Q}$, providing a more diverse and challenging set of contrasts for the query. The modified InfoNCE loss is given by:

$$\mathcal{L}(\mathbf{q}, \mathbf{k}, \mathcal{Q}, \mathcal{S}) = -\log \frac{\exp(\mathbf{q}^\top \cdot \mathbf{k}/\tau)}{\exp(\mathbf{q}^\top \cdot \mathbf{k}/\tau) + \sum_{\mathbf{n} \in \mathcal{Q}} \exp(\mathbf{q}^\top \cdot \mathbf{n}/\tau) + \sum_{\mathbf{s} \in \mathcal{S}} \exp(\mathbf{q}^\top \cdot \mathbf{s}/\tau)}. \tag{9}$$

Here, $\tau$ is the temperature parameter, $\mathcal{Q}$ is the set of original memory-based negatives, and $\mathcal{S}$ is the set of synthetic hard negatives. By incorporating both real and synthetic negatives, the model is exposed to a wider variety of challenging examples, which encourages learning more robust and generalizable representations. The overall computational overhead of SynCo is roughly equivalent to increasing the queue/memory by $\sum_{i=1}^{6} N_i \ll K$, along with the additional yet negligible cost of generating the synthetic negatives. Since synthetic negatives are generated "*on-the-fly*" during training and can be efficiently computed in parallel with the forward pass, the additional computational cost is marginal compared to the base contrastive learning framework. Moreover, the memory footprint remains manageable as synthetic negatives do not need to be stored persistently in the memory bank.

## 5 Experiments

In this section, we present comprehensive experiments demonstrating SynCo's effectiveness across multiple benchmarks, including ImageNet linear evaluation, semi-supervised learning, and transfer learning to object detection tasks.

### 5.1 Implementation Details

We pretrain SynCo on ImageNet ILSVRC-2012 (Deng et al., 2009) using a ResNet-50 encoder (He et al., 2015). Our method builds upon MoCo-v2 (Chen et al., 2020c); thus, it is only *fair* to compare against other MoCo-based methods (Chen et al., 2020c; Li et al., 2021a; Kalantidis et al., 2020; Yeh et al., 2022), which share similar architectures and training setups (see **bold** entries in Tables 1 to 5, indicating the best performance among MoCo-based methods). For training, unless stated otherwise, we use $K = 65$k. For SynCo, we also have a warm-up of 10 epochs, i.e. for the first epochs we do *not* synthesize hard negatives. We set SynCo's hyperparameters $\sigma$, $\delta$, and $\eta$ to 0.01. For hard negative generation, we use the top $N = 1024$ hardest negatives, with $N_1 = N_2 = N_3 = 256$ and $N_4 = N_5 = N_6 = 64$. For ImageNet linear evaluation, we train a linear classifier on frozen features for 100 epochs, using a batch size of 256 and a cosine learning rate schedule. Initial learning rates are set to 30.0 for ImageNet and 10.0 for ImageNet-100. To evaluate transfer learning, we apply SynCo to object detection tasks. For PASCAL VOC (Everingham et al., 2009), we fine-tune a Faster R-CNN (Ren et al., 2016) on `trainval07+12` and test on `test2007`. For COCO (Lin et al., 2015), we use a Mask R-CNN (He et al., 2018), fine-tuning on `train2017` and evaluating on `val2017`. We employ Detectron2 (Wu et al., 2019) and report standard AP metrics, following (He et al., 2020) *without* additional hyperparameter tuning. Detailed implementation details along ablations are provided in Appendices B and D.

### 5.2 Linear Evaluation on ImageNet

We evaluate the SynCo representation by training a linear classifier on top of the frozen features pretrained on ImageNet (details in Appendix B.2). With 200 epochs pretraining (Table 1), SynCo obtains 67.9% $\pm$ 0.16% top-1 accuracy and 88.0% $\pm$ 0.05% top-5 accuracy, showing strong improvements over MoCo-based methods (**+0.4%** over MoCo-v2, **+1.0%** over MoCHI, **+0.3%** over PCL-v2 and DCL). While MoCHI's hard negative generation leads to lower performance than MoCo-v2, our synthetic hard negatives achieve consistent gains. When training for 800 epochs (Table 2), SynCo reaches 70.7% top-1 accuracy (**+2.0%** over MoCHI) and 89.8% top-5 accuracy. However, at 800 epochs, it does not surpass MoCo-v2, similar to what is also observed with MoCHI, likely due to an overly hard proxy task (Kalantidis et al., 2020). As illustrated in Figure 5, we observe that the performance of standard SynCo begins to plateau around epoch 400, suggesting that continued generation of synthetic negatives may lead to an overly challenging proxy task in later stages of training. When we implement SynCo[‡] (stopping synthetic negative generation after epoch 400), the model achieves superior performance, reaching 71.6% top-1 accuracy—a **+0.5%** improvement over MoCo-v2.

### 5.3 Semi-supervised Training on ImageNet

We evaluate SynCo in a semi-supervised setting using 1% and 10% of labeled ImageNet data (details in Appendix B.3). Results in Table 3 show that with 1% labels, SynCo achieves 50.8% $\pm$ 0.21% top-1 accuracy (**+25.4%** over supervised baseline, **+2.6%** over MoCo-v2, **+2.5%** over SimCLR) and 77.5% $\pm$ 0.12% top-5

Table 1: Top-1 and top-5 accuracies (in %) under linear evaluation on ImageNet ILSVRC-2012 with 200 epochs of pretraining using ResNet-50. Result for SynCo are given as max over 3 runs.

| Method | Top-1 | Top-5 |
|---|---|---|
| *Supervised* | 76.5 | - |
| PIRL (Misra & van der Maaten, 2019) | 63.6 | - |
| LA (Zhuang et al., 2019) | 60.2 | - |
| InfoMin (Tian et al., 2020b) | 70.1 | 89.4 |
| SimSiam (Chen & He, 2020) | 68.1 | - |
| MSF (Koohpayegani et al., 2021) | 72.4 | - |
| ReSSL (Zheng et al., 2021) | 62.9 | - |
| SimCLR + DCL (Yeh et al., 2022) | 65.8 | - |
| SimCLR + DCLW (Yeh et al., 2022) | 66.9 | - |
| *MoCo-based* | | |
| MoCo (He et al., 2020) | 60.7 | - |
| PCL-v1 (Li et al., 2021a) | 61.5 | - |
| MoCo-v2 (Chen et al., 2020c) | 67.5 | 90.1 |
| PCL-v2 (Li et al., 2021a) | 67.6 ↑0.1 | - |
| MoCo-v2 + DCL (Yeh et al., 2022) | 67.6 ↑0.1 | - |
| MoCHI (Kalantidis et al., 2020) | 66.9 ↓0.6 | - |
| SynCo (ours) | **68.1** ↑0.6 | 88.0 |

Table 2: Top-1 and top-5 accuracies (in %) under linear evaluation on ImageNet ILSVRC-2012 for models trained with extended epochs using ResNet-50. Results for SynCo are based on 1 run.

| Method | Epochs | Top-1 | Top-5 |
|---|---|---|---|
| PIRL (Misra & van der Maaten, 2019) | 800 | 63.6 | - |
| InfoMin (Tian et al., 2020b) | 800 | 73.0 | 91.1 |
| SimSiam (Chen & He, 2020) | 800 | 68.1 | - |
| SimCLR (Chen et al., 2020b) | 1000 | 69.3 | - |
| BYOL (Grill et al., 2020) | 1000 | 74.3 | 91.6 |
| SwAV[†] (Caron et al., 2020) | 800 | 71.8 | - |
| SwAV (Caron et al., 2020) | 800 | 75.3 | - |
| Barlow Twins (Zbontar et al., 2021) | 1000 | 73.2 | 91.0 |
| VICReg (Bardes et al., 2022a) | 1000 | 73.2 | 91.1 |
| VICRegL (Bardes et al., 2022b) | 300 | 70.4 | - |
| *MoCo-based* | | | |
| MoCo-v2 (Chen et al., 2020c) | 800 | 71.1 | 90.1 |
| MoCHI (Kalantidis et al., 2020) | 800 | 68.7 ↓2.4 | - |
| SynCo (ours) | 800 | 70.7 ↓0.4 | 89.8 |
| SynCo[‡] (ours) | 800 | **71.6** ↑0.5 | 90.5 |

[†] Without multi-crop augmentation (by default in SwAV).
[‡] We stop generating synthetic negatives at epoch 400.

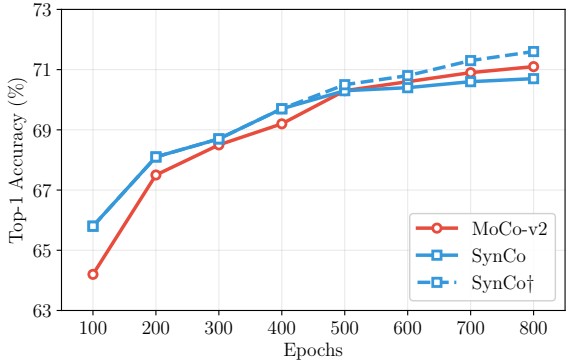

Figure 5: Top-1 accuracy progression on ImageNet linear evaluation comparing MoCo-v2, SynCo, and SynCo[‡] (stopping synthetic hard negative generation after epoch 400). Results show accuracy every 100 epochs during 800-epoch training.

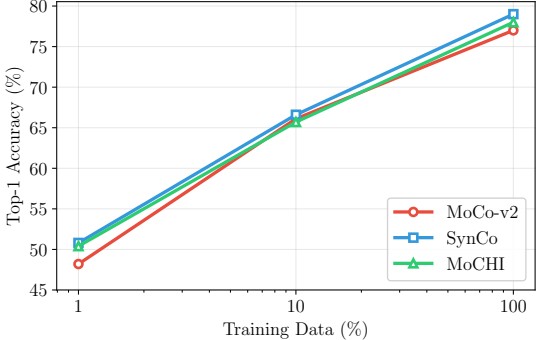

Figure 6: Semi-supervised learning performance comparison across different training data fractions (1%, 10%, and 100%) on ImageNet. SynCo consistently outperforms MoCo-v2 and MoCHI across all data regimes.

accuracy. With 10% labels, it reaches 66.6% ± 0.19% top-1 (**+10.2%** over supervised, **+0.5%** over MoCo-v2, **+1.0%** over SimCLR) and 88.0% ± 0.10% top-5 accuracy. Interestingly, when we stop generating synthetic negatives after epoch 200, similar to our observation in linear evaluation, performance improves further to 51.2% ± 0.23% top-1 and 78.0% ± 0.14% top-5 with 1% labels, and 67.1% ± 0.20% top-1 and 88.7% ± 0.11% top-5 with 10% labels. We also evaluate SynCo's performance when fine-tuning with 100% of the labeled ImageNet data. As shown in Figure 6, SynCo demonstrates consistent improvements over MoCo-based methods across all training data fractions. With the full dataset (100% labels), SynCo achieves 79.0% top-1 accuracy, outperforming MoCo-v2 (77.0%) by **+2.0%** and MoCHI (78.0%) by **+1.0%**. This comprehensive evaluation across 1%, 10%, and 100% of labeled data demonstrates that SynCo's synthetic hard negatives provide robust improvements regardless of the amount of available supervision, with particularly pronounced benefits in low-data regimes where the quality of learned representations becomes even more critical.

Table 3: Semi-supervised learning on ImageNet ILSVRC-2012 with 1% and 10% training examples using ResNet-50. Results for SynCo are averaged over 3 runs.

| Method | Epochs | Top-1 1% | Top-1 10% | Top-5 1% | Top-5 10% |
|---|---|---|---|---|---|
| *Supervised* | | 25.4 | 56.4 | 48.4 | 80.4 |
| InstDis (Wu et al., 2018) | 200 | - | - | 39.2 | 77.4 |
| SimCLR (Chen et al., 2020b) | 1000 | 48.3 | 65.6 | 75.5 | 87.8 |
| Barlow Twins (Zbontar et al., 2021) | 1000 | 55.0 | 69.7 | 79.2 | 89.3 |
| BYOL (Grill et al., 2020) | 1000 | 53.2 | 68.8 | 78.4 | 89.0 |
| SwAV (Caron et al., 2020) | 800 | 53.9 | 70.2 | 78.5 | 89.9 |
| PAWS (Assran et al., 2021) | 200 | 63.8 | 73.9 | - | - |
| *MoCo-v2-based* | | | | | |
| MoCo-v2 (repr.) | 800 | 48.2 | 66.1 | 75.8 | 87.6 |
| MoCHI (repr.) | 800 | 50.4 | 65.7 | 76.2 | 87.2 |
| SynCo (ours) | 800 | 50.8 | 66.6 | 77.5 | 88.0 |
| SynCo[‡] (ours) | 800 | **51.2** | **67.1** | **78.0** | **88.7** |

[‡] We stop generating synthetic negatives at epoch 400.

Table 4: Transfer learning on PASCAL VOC07+12 using R50-C4. We report AP, $AP_{50}$, and $AP_{75}$, which are standard COCO metrics. Results for SynCo are averaged over 3 runs.

| Method | Epochs | $AP$ | $AP_{50}$ | $AP_{75}$ |
|---|---|---|---|---|
| *Supervised* | 200 | 53.5 | 81.3 | 58.8 |
| *Random init* | 200 | 33.8 | 60.2 | 33.1 |
| InfoMin (Tian et al., 2020b) | 200 | 57.6 | 82.7 | 64.6 |
| SimSiam (Chen & He, 2020) | 200 | 57.0 | 82.4 | 63.7 |
| BYOL (Grill et al., 2020) | 300 | 51.9 | 81.0 | 56.5 |
| SwAV (Caron et al., 2020) | 800 | 56.1 | 82.6 | 62.7 |
| Barlow Twins (Zbontar et al., 2021) | 1000 | 56.8 | 82.6 | 63.4 |
| SimCLR (Chen et al., 2020b) | 1000 | 56.3 | 81.9 | 62.5 |
| *Detection-specific* | | | | |
| SoCo (Wei et al., 2021) | 100 | 59.1 | 83.4 | 65.6 |
| InsLoc (Yang et al., 2021) | 200 | 57.9 | 82.9 | 64.9 |
| DetCo (Xie et al., 2021a) | 200 | 57.8 | 82.6 | 64.2 |
| ReSim (Xiao et al., 2021) | 200 | 58.7 | 83.1 | 66.3 |
| *MoCo-based* | | | | |
| MoCo (He et al., 2020) | 200 | 55.9 | 81.5 | 62.6 |
| MoCo-v2 (Chen et al., 2020c) | 200 | 57.0 | 82.4 | 63.6 |
| MoCHI (Kalantidis et al., 2020) | 200 | **57.5** | **82.7** | **64.4** |
| SynCo (ours) | 200 | 57.2 | 82.6 | 63.9 |

Table 5: Transfer learning on COCO using R50-C4. Both 1× and 2× training schedules are reported. We report AP, $AP_{50}$, and $AP_{75}$. *bb* denotes bounding box detection, and *msk* denotes instance segmentation. Results for SynCo are averaged over 3 runs.

| Method | Epochs | COCO 1× schedule $AP^{bb}$ | $AP_{50}^{bb}$ | $AP_{75}^{bb}$ | $AP^{msk}$ | $AP_{50}^{msk}$ | $AP_{75}^{msk}$ | COCO 2× schedule $AP^{bb}$ | $AP_{50}^{bb}$ | $AP_{75}^{bb}$ | $AP^{msk}$ | $AP_{50}^{msk}$ | $AP_{75}^{msk}$ |
|---|---|---|---|---|---|---|---|---|---|---|---|---|---|
| *Supervised* | 200 | 38.2 | 58.2 | 41.2 | 33.3 | 54.7 | 35.2 | 40.0 | 59.9 | 43.1 | 34.7 | 56.5 | 36.9 |
| *Random init* | 200 | 26.4 | 44.0 | 27.8 | 29.3 | 46.9 | 30.8 | 35.6 | 54.6 | 38.2 | 31.4 | 51.5 | 33.5 |
| InfoMin (Tian et al., 2020b) | 200 | 39.0 | 58.5 | 42.0 | 34.1 | 55.2 | 36.3 | 41.3 | 61.2 | 45.0 | 36.0 | 57.9 | 38.3 |
| SimSiam (Chen & He, 2020) | 200 | 39.2 | 59.3 | 42.1 | 34.4 | 56.0 | 36.7 | - | - | - | - | - | - |
| BYOL (Grill et al., 2020) | 300 | - | - | - | - | - | - | 40.3 | 60.5 | 43.9 | 35.1 | 56.8 | 37.3 |
| SwAV (Caron et al., 2020) | 800 | 38.4 | 58.6 | 41.3 | 33.8 | 55.2 | 35.9 | - | - | - | - | - | - |
| Barlow Twins (Zbontar et al., 2021) | 1000 | 39.2 | 59.0 | 42.5 | 34.3 | 56.0 | 36.5 | - | - | - | - | - | - |
| SimCLR (Chen et al., 2020b) | 1000 | - | - | - | - | - | - | 40.3 | 60.5 | 43.9 | 35.1 | 56.8 | 37.3 |
| *Detection-specific* | | | | | | | | | | | | | |
| SoCo (Wei et al., 2021) | 100 | 40.4 | 60.4 | 43.7 | 34.9 | 56.8 | 37.0 | 41.1 | 61.0 | 44.4 | 35.6 | 57.5 | 38.0 |
| InsLoc (Yang et al., 2021) | 200 | 39.5 | 59.1 | 42.7 | 34.5 | 56.0 | 36.8 | 41.4 | 60.9 | 45.0 | 35.9 | 57.6 | 38.4 |
| DetCo (Xie et al., 2021a) | 200 | 39.8 | 59.7 | 43.0 | 34.7 | 56.3 | 36.7 | 41.3 | 61.2 | 45.0 | 35.8 | 57.9 | 38.2 |
| ReSim (Xiao et al., 2021) | 200 | 39.7 | 59.0 | 43.0 | 34.6 | 55.9 | 37.1 | - | - | - | - | - | - |
| *MoCo-based* | | | | | | | | | | | | | |
| MoCo (He et al., 2020) | 200 | 38.5 | 58.3 | 41.6 | 33.6 | 54.8 | 35.6 | 40.7 | 60.5 | 44.1 | 35.4 | 57.3 | 37.6 |
| MoCo-v2 (Chen et al., 2020c) | 200 | 38.9 | 58.4 | 42.0 | 34.2 | 55.2 | 36.5 | 40.7 | 60.5 | 44.1 | 35.6 | **57.4** | 37.1 |
| MoCHI (Kalantidis et al., 2020) | 200 | 39.2 | 58.9 | 42.4 | 34.3 | 55.5 | 36.6 | - | - | - | - | - | - |
| SynCo (ours) | 200 | **39.9** | **59.6** | **43.3** | **34.9** | **56.5** | **36.9** | 41.0 | 60.6 | 44.8 | 35.7 | 57.4 | 38.1 |

## 5.4 Transferring to Detection

We evaluate the SynCo representation, pretrained for 200 epochs, by applying it to detection tasks (details in Appendix B.4). Table 4 shows that on PASCAL VOC, SynCo achieves strong results (57.2 AP) comparable to MoCHI (57.5 AP), while significantly outperforming the supervised baseline (**+3.7** AP). On the more challenging COCO dataset (Table 5) with 1× schedule, SynCo shows consistent improvements over the supervised baseline ($AP^{bb}$ **+1.7**, $AP^{msk}$ **+1.6**) and MoCo-v2 ($AP^{bb}$ **+1.0**, $AP^{msk}$ **+0.8**). SynCo achieves competitive performance with detection-specific methods, showing comparable results to DetCo (39.8 vs 39.9 $AP^{bb}$) and InsLoc (39.5 vs 39.9 $AP^{bb}$), despite using a general contrastive learning framework. Additional results are provided in Appendix C.1.

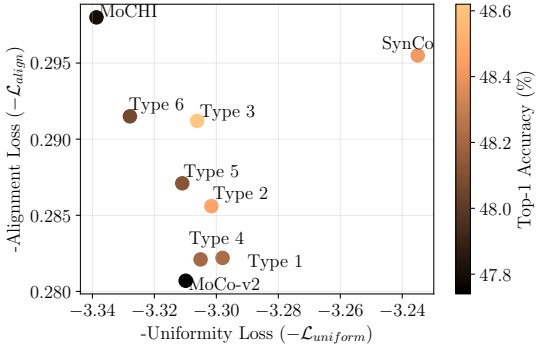

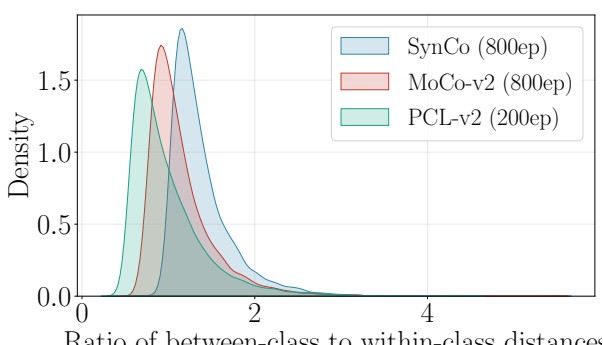

Figure 7: Performance comparison of MoCo-v2, MoCHI, and SynCo (under various configurations) on ImageNet-100 in terms of alignment and uniformity metrics. The x-axis and y-axis represent $-\mathcal{L}_{\text{uniform}}$ and $-\mathcal{L}_{\text{align}}$, respectively. The model with the highest performance is located in the upper-right corner of the chart.

Figure 8: Distribution of the ratio between inter-class and intra-class distances for MoCo-based methods. Higher values indicate better class separation. For clarity, we only show MoCo-v2 (800 epochs), PCL-v2 (200 epochs), and SynCo (800 epochs).

## 6  Discussion

This section examines how synthetic negatives affect proxy task difficulty and shape representation space utilization.

### 6.1  Is the Proxy Task More Difficult?

Figure 3b depicts the proxy task performance for different configurations of SynCo. We observe that incorporating synthetic negatives leads to faster learning and improved performance. Each type of synthetic negative accelerates learning compared to the MoCo-v2 baseline, with the full SynCo configuration showing the most significant improvement (see Table 15) and the lowest final proxy task performance. This indicates that SynCo presents the most challenging proxy task. This is evidenced by $\max \ell(\mathbf{s}_k^i) > \max \ell(\mathbf{n}_j)$, where $\mathbf{s}_k^i \in S^i$ are synthetic negatives and $\mathbf{n}_j \in \hat{\mathcal{Q}}_N$ are original negatives. Through SynCo, we modulate proxy task difficulty via synthetic negatives, pushing the model to learn more robust features.

### 6.2  Evaluating the Usage of the Representation Space

To assess learned representations, we employ alignment and uniformity metrics (Wang & Isola, 2020) (details in Appendix B.5). These metrics provide insights into representation space utilization, with alignment quantifying the grouping of similar samples and uniformity measuring representation spread across the hypersphere. Figure 7 presents results for various MoCo-based methods. Our findings demonstrate that SynCo significantly enhances the uniformity of representations compared to MoCo-v2 and MoCHI, demonstrating improved utilization of the representation space in the proxy task. Furthermore, the incorporation of synthetic negatives (types 1 to 6) leads to improved alignment. These results suggest that SynCo's approach to synthetic negative generation and contrastive learning yields stronger and more well-distributed feature representations.

### 6.3  Class Concentration Analysis

To quantify the structure of the learned latent space, we examine the relationship between within-class and between-class distances. Figure 8 shows the distribution of ratios between inter-class and intra-class $\ell_2$-distances for representations learned by various MoCo-based contrastive methods on the ImageNet validation set. A higher mean ratio indicates that representations are better concentrated within classes while maintaining greater separation between classes, reflecting improved linear separability (aligned with Fisher's

linear discriminant analysis principles (Friedman et al., 2009)). After 800 training epochs, SynCo achieves a mean ratio of 1.384, significantly surpassing MoCo-v2 (1.146) and PCL-v2 (0.988). Additional results are provided in Appendix C.2.

## 7 Conclusion

This paper introduces SynCo, a novel contrastive learning approach leveraging synthetic hard negatives to enhance visual representation learning. By generating diverse and challenging negatives "*on-the-fly*", SynCo overcomes the limitations of maintaining an effective hard negative pool throughout training. Experiments demonstrate that SynCo accelerates learning and produces more robust, transferable representations.

While our experiments primarily employed the MoCo framework for the lower batch size requirements, the proposed hard negative generation strategies are general and applicable to any contrastive learning method that benefits from hard negatives, such as SimCLR (Chen et al., 2020b), CPC (van den Oord et al., 2019), PIRL (Misra & van der Maaten, 2019), and other (Wang & Qi, 2022; Dwibedi et al., 2021; Tian et al., 2021). These methods, which utilize the InfoNCE loss function (or its variants (Chen et al., 2020b; Dwibedi et al., 2021)) and instance discrimination as the pretext task, gain from SynCo's enhanced hard negative generation. While more complex methods (Caron et al., 2020; Grill et al., 2020) use additional tricks to refine representation learning, SynCo shows strong improvements within MoCo-v2's simpler framework. By introducing synthetic hard negatives, these methods access more challenging, informative contrasts, potentially improving feature representations. Furthermore, SynCo's applicability extends beyond visual representation learning, offering benefits in domains such as natural language processing, audio processing, and other areas where contrastive learning is relevant.

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

# Appendix

## Contents

## A   Algorithm

Algorithm 1 provides the pseudo-code of SynCo.

## B   Implementation Details

We implement SynCo in PyTorch following the implementation of MoCo[1]. Specifically, we follow the same setting as MoCo-v2.

### B.1   Pretraining

**Datasets.**  We evaluate the proposed method on ImageNet ILSVRC-2012[2] (Deng et al., 2009), which includes 1000 classes and is commonly used in previous self-supervised methods (Chen et al., 2020b; Chen & He, 2020; Zbontar et al., 2021; Zhang et al., 2022b). The dataset consists of 1.28 million training images and 50,000 validation images. We also conduct ablation studies on ImageNet-100 (Khosla et al., 2021), a subset of 100 classes derived from ImageNet, with 126,689 training images and 5,000 validation images. Both

---

[1]Available at: `https://github.com/facebookresearch/moco`.
[2]Available at: `https://www.image-net.org/`.

---

**Algorithm 1** Pseudocode of SynCo in a PyTorch-like style.

---

```python
# f_q, f_k: encoder networks for query and key
# queue: dictionary as a queue of K keys (CxK)
# m: momentum
# t: temperature
# hard_neg_functions: list of functions to generate
# synthetic negatives (type 1 to 6)

f_k.params = f_q.params # initialize
for x in loader: # load a minibatch x with N samples
    x_q = aug(x) # a randomly augmented version
    x_k = aug(x) # another randomly augmented version

    q = f_q.forward(x_q) # queries: NxC
    k = f_k.forward(x_k) # keys: NxC
    k = k.detach() # no gradient to keys

    # positive logits: Nx1
    l_pos = bmm(q.view(N,1,C), k.view(N,C,1))

    # negative logits: NxK
    l_neg = mm(q.view(N,C), queue.view(C,K))

    # find indices of the top-(N_hard) hard negatives
    idxs_hard = topk(l_neg, k=N_hard)

    # generate hard negatives
    for func in hard_neg_functions:
        # generate hard negatives of type i
        s_neg = func(q, queue, idxs_hard)
        # compute logits for synthetic negatives
        l_syn = bmm(q.view(N,C), s_neg.view(N,C))
        # append hard negatives logits
        l_neg = cat([l_neg, l_syn], dim=1)

    # logits: Nx(1+K+N_hard)
    logits = cat([l_pos, l_neg], dim=1)

    # contrastive loss, positives are the 0-th
    labels = zeros(len(logits))
    loss = CrossEntropyLoss(logits/t, labels)

    # SGD update: query network
    loss.backward()
    update(f_q.params)

    # momentum update: key network
    f_k.params = m*f_k.params+(1-m)*f_q.params

    # update dictionary
    enqueue(queue, k) # enqueue the current minibatch
    dequeue(queue) # dequeue the earliest minibatch
```

---

`bmm`: batch matrix multiplication; `mm`: matrix multiplication; `cat`: concatenation; `topk`: returns the indices of the top-k values.

datasets are well-balanced in class distribution, and the images contain iconic views of objects, as is common in vision tasks (He et al., 2015; Zbontar et al., 2021).

**Augmentation.** Each input image is transformed twice to generate two different views. For SynCo, we use the same augmentation as used in (Chen et al., 2020c) and (Kalantidis et al., 2020) for a fair comparison. We transform each input image with two sampled augmentations to produce two distorted versions of the input. The augmentation pipeline consists of random cropping, resizing to $224 \times 224$, randomly flipping the images horizontally, applying color distortion, optionally converting to grayscale, adding Gaussian blurring.

**Architecture.** Both the encoder $f_q$ and $f_k$ consist of a backbone and a projection head. The encoder $f_k$ is updated by the moving average of $f_q$. As our base encoder, we adopt ResNet-50 (2048 output units). The projection head is a 2-layer MLP, following (Chen et al., 2020c): the hidden layers of the MLP are 2048-d and are with ReLU (Nair & Hinton, 2010); the output layer of the MLP is 128-d, without ReLU.

**Optimization.** We follow the same setting as (Chen et al., 2020c). We utilize the SGD optimizer (Ruder, 2017) with a base learning rate of 0.03 ($= 0.03 \times$ batch_size$/256$), where we scale the learning rate with the batch size as in (Chen et al., 2020b), and a weight decay of $10^{-4}$. The training schedule begins with a warm-up period during the first 10 epochs in which the learning rate linearly increases from 0 to the base

learning rate. Following this, the learning rate gradually decreases to zero following a cosine decay schedule without restarts. The batch size for ImageNet is set to 256 distributed over 4 NVIDIA L40 GPUs. The total training duration is set to 200/800 epochs for ImageNet. For pretraining, SynCo takes approximately 43 hours (1.8 days) and 8 kWh of power for 100 epochs.

**Hyperparameters.** We empirically set SynCo's hyperparameters to $\sigma = 0.01$, $\delta = 0.01$, and $\eta = 0.01$. A thorough analysis of these hyperparameters revealed no significant difference in performance when these values are varied within reasonable bounds (also see Appendix D), indicating that our method is robust to a range of practical settings. For hard negative generation, we select the top $N = 1024$ hardest negatives and set $N_1 = N_2 = N_3 = 256$ and $N_4 = N_5 = N_6 = 64$ to maintain a balanced total number of generated hard negatives. A detailed analysis of the choice of $N_i$, $i = 1, \ldots, 6$ is provided in Appendix D. We tested various similarity functions, including cosine similarity, Euclidean, and Mahalanobis distances, for generating gradient-based synthetic hard negatives. Our results revealed no significant differences in model performance across these similarity measures. Therefore, we opted to use the dot product similarity function, which simplifies computation and aligns with the InfoNCE loss used in SynCo's contrastive learning framework.

For detailed configuration of SynCo pretraining, including architecture and optimization parameters, see Table 6.

Table 6: Architecture and optimization hyperparameters used for SynCo pretraining.

| Parameter | Value |
|---|---|
| *Architecture* | |
| Backbone | ResNet-50 |
| Projection head | 2-layer MLP |
| Projection head activation | ReLU |
| *Optimization* | |
| Optimizer | SGD |
| Momentum | 0.9 |
| Base learning rate | 0.03 |
| Weight decay | $10^{-4}$ |
| Warm-up | 10 epochs |
| Batch size | 256 |
| Training epochs | 200/800 epochs |
| Training time | $\sim$43 hours/100 epochs |
| *MoCo* | |
| Queue size $K$ | 65536 |
| Momentum $m$ | 0.999 |
| Temperature $\tau$ | 0.2 |
| *SynCo* | |
| Hardest negatives $N$ | 1024 |
| Synthetic $N_i$, $i = 1, 2, 3$ | 256 |
| Synthetic $N_i$, $i = 4, 5, 6$ | 64 |
| Hyperparameters $\sigma, \delta, \eta$ | 0.01 |

## B.2 Linear Evaluation

We follow the linear evaluation protocol of (He et al., 2020) and as in (Kornblith et al., 2019; Kolesnikov et al., 2019; Chen et al., 2020b; Grill et al., 2020; van den Oord et al., 2019), which consists in training a linear classifier on top of the frozen representation, i.e., without updating the network parameters nor the batch statistics. At training time, we apply spatial augmentations, i.e., random crops with resize to $224 \times 224$ pixels, and random flips. At test time, images are resized to 256 pixels along the shorter side using bicubic resampling, after which a $224 \times 224$ center crop is applied. In both cases, we normalize the color channels by subtracting the average color and dividing by the standard deviation, after applying the augmentations. We

optimize the cross-entropy loss using SGD with Nesterov momentum over 100 epochs, using a batch size of 256 and a momentum of 0.9. We do not use any regularization methods such as weight decay, gradient clipping (Cubuk et al., 2019), tclip (Bachman et al., 2019), or logits regularization. We use a learning rate of 30.0 for ImageNet ILSVRC-2012 and 10.0 for ImageNet-100. We train using 4 NVIDIA L40 GPUs.

### B.3 Semi-supervised Training

We follow the semi-supervised learning protocol of (Chen et al., 2020b; Kornblith et al., 2019; Zhai et al., 2020; Grill et al., 2020). We first initialize the network with the parameters of the pretrained representation, and fine-tune it with a subset of ImageNet ILSVRC-2012 labels. At training time, we apply spatial augmentations, i.e., random crops with resize to $224\times 224$ pixels and random flips. At test time, images are resized to 256 pixels along the shorter side using bicubic resampling, after which a $224 \times 224$ center crop is applied. In both cases, we normalize the color channels by subtracting the average color and dividing by the standard deviation (computed on ImageNet), after applying the augmentations. We optimize the cross-entropy loss using SGD with Nesterov momentum. We used a batch size of 1024, a momentum of 0.9. We do not use any regularization methods such as weight decay, gradient clipping (Cubuk et al., 2019), tclip (Bachman et al., 2019), or logits rescaling. Similar to (Caron et al., 2020), we sweep over the learning rates $\{0.01, 0.02, 0.05, 0.1, 0.005\}$ and the number of epochs $\{30, 60\}$. We train using 4 NVIDIA L40 GPUs.

### B.4 Object Detection

We follow the object detection protocol of (He et al., 2020; Chen et al., 2020c). We first initialize the network with the parameters of the pretrained representation, and fine-tune it on PASCAL VOC (Everingham et al., 2009)[3] and COCO (Lin et al., 2015)[4] datasets. During training, we apply spatial augmentations, specifically random resizing and random horizontal flipping. During testing, images are resized to a fixed size of 800 pixels along the shorter side. The R50-C4 backbones, similar to those used in Detectron2 (Wu et al., 2019), conclude at the `conv4` stage. Subsequently, the box prediction head is composed of the `conv5` stage, which includes global pooling, followed by a BN layer. We train using 8 NVIDIA RTX 6000 GPUs.

**PASCAL VOC object detection.** We use a Faster R-CNN (Ren et al., 2016) with the SGD optimizer at a base learning rate of 0.02, a momentum of 0.1, and a weight decay of 0.0001, and a batch size of 16. The model is trained for 24,000 iterations using a step learning rate scheduler, where the learning rate is reduced at 18,000 and 22,000 iterations. Images are scaled to $480 \times 800$ pixels during training and resized to 800 pixels on the longer side for inference.

**COCO object detection.** We use a Mask R-CNN (He et al., 2018) with the SGD optimizer at a base learning rate of 0.02, a momentum of 0.1, and a weight decay of 0.0001, and a batch size of 16. For the $1\times$ schedule, the model trains for 90,000 iterations with learning rate reductions at 60,000 and 80,000 iterations. For the $2\times$ schedule, it trains for 180,000 iterations with learning rate reductions at 120,000 and 160,000 iterations. A warm-up period is applied for the first 100 iterations. Images are resized to $640 \times 800$ pixels during training and normalized to 800 pixels on the longer side for inference.

### B.5 Alignment and Uniformity

We follow the protocol of (Kalantidis et al., 2020) but training the network 100 epochs on ImageNet-100. We calculate the alignment and uniformity based on (Wang & Isola, 2020). The alignment loss $\mathcal{L}_{\text{align}}$ and uniformity loss $\mathcal{L}_{\text{uniform}}$ are computed as follows:

$$\mathcal{L}_{\text{align}}(\mathbf{x}, \mathbf{y}) = \mathbb{E}_{(\mathbf{x},\mathbf{y})\sim p_{\text{data}}} \left[ \|f_q(\mathbf{x}) - f_k(\mathbf{y})\|_2^\alpha \right] \tag{10}$$

$$\mathcal{L}_{\text{uniform}}(\mathbf{x}) = \log \mathbb{E}_{\mathbf{x},\mathbf{y}\sim p_{\text{data}}} \left[ \exp(-t\|f_q(\mathbf{x}) - f_k(\mathbf{y})\|_2^2) \right] \tag{11}$$

---

[3]Available at `https://host.robots.ox.ac.uk/pascal/VOC/`.

[4]Available at `https://cocodataset.org/`.

where $\mathbf{x}$ and $\mathbf{y}$ is a pair of positive images, $\alpha$ is a hyperparameter typically set to 2, and $t$ controls the sharpness of the distribution, also set to 2. Here, $p_{\text{data}}$ represents the empirical distribution of the data, from which pairs of embeddings $(\mathbf{x}, \mathbf{y})$ are sampled. We implement these losses in PyTorch following the original implementation[5].

## B.6 ImageNet-100 Subsets

The list of classes from ImageNet-100[6] is randomly sampled from the original ImageNet ILSVRC-2012 dataset and is the same as that used in (Tian et al., 2020a). The list is shown in Table 7.

Table 7: The list of classes from ImageNet-100, randomly sampled from ImageNet ILSVRC-2012.

| ImageNet-100 |
| --- |
| n02869837 n01749939 n02488291 n02107142 n13037406 n02091831 n04517823 n04589890 n03062245 n01773797 |
| n01735189 n07831146 n07753275 n03085013 n04485082 n02105505 n01983481 n02788148 n03530642 n04435653 |
| n02086910 n02859443 n13040303 n03594734 n02085620 n02099849 n01558993 n04493381 n02109047 n04111531 |
| n02877765 n04429376 n02009229 n01978455 n02106550 n01820546 n01692333 n07714571 n02974003 n02114855 |
| n03785016 n03764736 n03775546 n02087046 n07836838 n04099969 n04592741 n03891251 n02701002 n03379051 |
| n02259212 n07715103 n03947888 n04026417 n02326432 n03637318 n01980166 n02113799 n02086240 n03903868 |
| n02483362 n04127249 n02089973 n03017168 n02093428 n02804414 n02396427 n04418357 n02172182 n01729322 |
| n02113978 n03787032 n02089867 n02119022 n03777754 n04238763 n02231487 n03032252 n02138441 n02104029 |
| n03837869 n03494278 n04136333 n03794056 n03492542 n02018207 n04067472 n03930630 n03584829 n02123045 |
| n04229816 n02100583 n03642806 n04336792 n03259280 n02116738 n02108089 n03424325 n01855672 n02090622 |

## B.7 Image Augmentations

During self-supervised training, SynCo uses the same augmentation as (Chen et al., 2020c). The augmentation parameters are detailed in Table 8.

Table 8: Parameters used to generate image augmentations.

| Parameter | MoCo-v2 $\mathcal{T}$ |
| --- | --- |
| Random crop probability | 1.0 |
| Horizontal flip probability | 0.5 |
| Vertical flip probability | 0.8 |
| Brightness adjustment max intensity | 0.4 |
| Contrast adjustment max intensity | 0.4 |
| Saturation adjustment max intensity | 0.2 |
| Hue adjustment max intensity | 0.1 |
| Color dropping probability | 0.2 |
| Gaussian blurring probability | 0.5 |
| Solarization probability | 0.0 |

# C  Additional Results

In this section we present extended results starting with object detection on PASCAL VOC, where SynCo demonstrates faster training and matches MoCo-v2's performance at 800 epochs. We then analyze the representations learned by SynCo through multiple perspectives. We examine the model's feature space using class concentration metrics, dimensionality reduction techniques (t-SNE, UMAP), and nearest neighbor analysis to understand its semantic organization. We investigate the robustness of learned representations under distribution shifts (ImageNet variants), corruptions, and adversarial attacks to assess their generalization capabilities. Through GradCAM visualizations, we also provide insights into which image regions contribute

---

[5]Available at: `https://github.com/Ssnl/align_uniform`.

[6]Available at: `https://github.com/HobbitLong/CMC/blob/master/imagenet100.txt`.

most to the model's feature extraction. These analyses collectively demonstrate SynCo's ability to learn discriminative and robust visual representations.

## C.1 Transferring to Detection

We evaluate the SynCo representation using a pretrained ResNet-50 model trained for 800 epochs on VOC dataset. The results are shown in Table 9. SynCo demonstrates faster training, achieving better results at lower epochs compared to MoCo-v2. At 200 epochs, SynCo already surpasses MoCo-v2 in terms of $AP_{50}$ and $AP_{75}$. However, when training is extended to 800 epochs, MoCo-v2 and SynCo perform on par, with both methods reaching similar performance.

Table 9: Results for object detection on PASCAL VOC. The values in bold indicate the maximum of each column.

| Method | Epochs | $AP$ | $AP_{50}$ | $AP_{75}$ |
|---|---|---|---|---|
| *Supervised* | 90 | 53.5 | 81.3 | 58.8 |
| MoCo (He et al., 2020) | 200 | 55.9 | 81.5 | 62.6 |
| MoCo-v2 (Chen et al., 2020c) | 200 | 57.0 | 82.4 | 63.6 |
| MoCo-v2 (Chen et al., 2020c) | 800 | **57.4** | 82.5 | **64.0** |
| SynCo (ours) | 200 | 57.2 | 82.6 | 63.9 |
| SynCo (ours) | 800 | **57.4** | **82.8** | **64.0** |

## C.2 Class Concentration Analysis

To quantify the overall structure of the learned latent space, we examine the relationship between within-class and between-class distances. Figure 9 compares the distribution of ratios between inter-class and intra-class $\ell_2$-distances of representations learned by different MoCo-based contrastive learning methods on the ImageNet validation set. A higher mean ratio indicates that the representations are better concentrated within their corresponding classes while maintaining better separation between different classes, suggesting improved linear separability (following Fisher's linear discriminant analysis principles (Friedman et al., 2009)).

Table 10: Statistical summary of the ratio between inter-class and intra-class distances for different MoCo-based methods. ↑ indicates higher is better, ↓ indicates lower is better. Higher mean indicates better class separation while lower standard deviation suggests more consistent feature learning across different classes.

| Method | Epochs | Mean ↑ | Median ↑ | Std ↓ |
|---|---|---|---|---|
| *Supervised* | 90 | 1.381 | **1.369** | **0.110** |
| MoCo (He et al., 2020) | 200 | 1.012 | 0.999 | 0.115 |
| MoCo-v2 (Chen et al., 2020c) | 200 | 1.061 | 0.971 | 0.358 |
| MoCo-v2 (Chen et al., 2020c) | 800 | 1.146 | 1.043 | 0.375 |
| PCL-v1 (Li et al., 2021a) | 200 | 0.930 | 0.869 | 0.312 |
| PCL-v2 (Li et al., 2021a) | 200 | 0.988 | 0.866 | 0.419 |
| SynCo (ours) | 200 | 1.104 | 1.001 | 0.383 |
| SynCo (ours) | 800 | **1.384** | 1.282 | 0.361 |

As shown in Table 10, SynCo trained for 800 epochs achieves the highest mean ratio (1.384) among all MoCo-based methods, approaching and slightly surpassing the supervised baseline (1.381). A higher mean ratio indicates better class separability, which is crucial for downstream classification tasks. This superior performance can be attributed to SynCo's synthetic hard negative generation strategies, which help create more discriminative feature representations.

The standard deviation of the ratio distribution provides insight into the consistency of learned features across different classes. Lower standard deviation suggests more uniform feature learning across all classes. While the supervised baseline achieves the lowest standard deviation (0.110), among MoCo-based methods, MoCo shows comparable consistency (0.115).

Notably, both SynCo variants (200 and 800 epochs) consistently outperform their MoCo-v2 counterparts at equivalent training epochs in terms of mean ratio (1.104 vs 1.061 at 200 epochs, and 1.384 vs 1.146 at 800 epochs), demonstrating the effectiveness of synthetic hard negatives in learning more discriminative features. The improvement in class concentration metrics aligns with SynCo's superior performance on downstream tasks, particularly in scenarios requiring fine-grained discrimination between similar classes. By focusing exclusively on methods built upon the MoCo framework, this comparison ensures a fair evaluation of SynCo's contributions to contrastive learning.

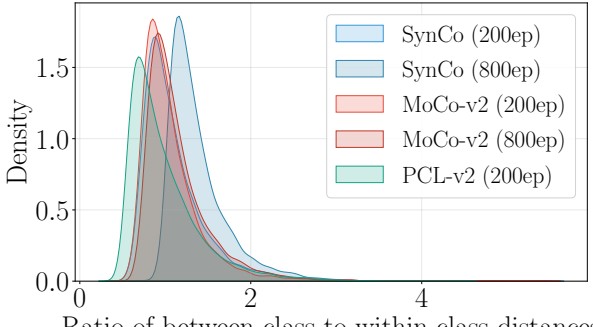

Figure 9: Distribution of the ratio between inter-class and intra-class distances for different MoCo-based methods. Higher values indicate better class separation. We show MoCo (200 epochs) (He et al., 2020), MoCo-v2 (Chen et al., 2020c) (200 and 800 epochs), PCL-v1 and PCL-v2 (Li et al., 2021a) (200 epochs), and SynCo (200 and 800 epochs).

## C.3 Robustness and Out-of-Distribution Evaluation

We evaluate the robustness and out-of-distribution (OOD) generalization capabilities of SynCo representations. For robustness evaluation, we employ four datasets: ImageNet-v2 (Recht et al., 2019), which comprises three sets of 10,000 images (matched frequency, threshold 0.7, and top images); ImageNet-A (Hendrycks et al., 2021b), which contains naturally adversarial examples; ImageNet-P (Hendrycks & Dietterich, 2019), which evaluates prediction stability under perturbations; and ImageNet-C (Hendrycks & Dietterich, 2019), which consists of 15 synthetically generated corruptions (e.g., blur, noise, weather);

For OOD generalization, we examine performance on five datasets: ImageNet-Sketch (Wang et al., 2019), containing 50,000 black-and-white sketches; ImageNet-R (Hendrycks et al., 2021a), consisting of 30,000 artistic renditions; ImageNet-O (Hendrycks et al., 2021b), designed for anomaly detection (evaluated using FPR95); ImageNet-Watermark (Li et al., 2023), testing robustness to watermark perturbations.

On all datasets, we evaluate the representations of a standard ResNet-50 encoder under a linear evaluation protocol, where we freeze the pretrained representations and train a linear classifier using the labeled ImageNet training set. The test evaluation is performed zero-shot, i.e., no training is done on the above datasets.

Table 11: Top-1 accuracy (in %) across different ImageNet variants using ResNet-50 as the backbone, except ImageNet-O (IN-O) where we evaluate using FPR95. IN: ImageNet; MF/T07/TI: ImageNet-v2 variants; IN-C: ImageNet-C; IN-A: ImageNet-A; IN-S: ImageNet-Sketch; IN-R: ImageNet-R; IN-W: ImageNet-Watermark. Results for SimCLR are from (Tomasev et al., 2022). We reproduce MoCo and MoCo-v2 linear probing since no checkpoints are available (thus results may differ from original implementation).

| Method | Epochs | IN | Robustness | | | | | | Out-Of-Distribution | | |
| | | | MF | T-0.7 | TI | IN-C | IN-A | IN-W | IN-S | IN-R | IN-O |
|---|---|---|---|---|---|---|---|---|---|---|---|
| *Supervised* | 90 | 76.1 | 63.1 | 72.3 | 77.6 | 39.8 | 0.0 | 48.7 | 24.1 | 36.2 | 81.4 |
| SimCLR (Chen et al., 2020b) | 1000 | 69.3 | 53.2 | 61.7 | 68.0 | 31.1 | - | - | 3.9 | 18.3 | - |
| MoCo (He et al., 2020) | 200 | 60.9 | 45.9 | 53.8 | 60.4 | 33.8 | 2.5 | 38.5 | 10.2 | 18.2 | 85.9 |
| MoCo-v2 (Chen et al., 2020c) | 200 | 67.8 | 54.8 | 63.0 | 69.0 | 51.4 | 2.8 | 44.2 | 17.5 | 27.8 | 81.9 |
| MoCo-v2 (Chen et al., 2020c) | 800 | 71.1 | 58.5 | 66.6 | 73.0 | 55.8 | 4.1 | 35.0 | 19.2 | 29.7 | 79.0 |
| SynCo (ours) | 200 | 68.1 | 54.9 | 63.7 | 69.8 | 51.6 | 3.2 | 42.8 | 16.5 | 26.7 | 82.5 |
| SynCo (ours) | 800 | 70.7 | 58.1 | 66.4 | 72.5 | 55.9 | 4.2 | 41.6 | 19.2 | 28.7 | 79.5 |

As shown in Table 11, SynCo demonstrates strong robustness across various distribution shifts, outperforming MoCo and SimCLR in most robustness benchmarks. At 200 epochs, SynCo achieves better results than MoCo and is competitive with MoCo-v2, particularly on ImageNet-C (51.6% top-1 accuracy) and ImageNet-A (3.2% top-1 accuracy). At 800 epochs, SynCo achieves comparable performance to MoCo-v2 across robustness

Table 12: Top-1 accuracy (%) for ImageNet-C corruption results using ResNet-50: noise (gaussian, shot, impulse), blur (defocus, glass, motion, zoom), weather (frost, snow, fog, brightness), digital (contrast, elastic, pixelate, jpeg). We reproduce MoCo and MoCo-v2 linear probing since no checkpoints are available.

| Method | Epochs | Noise | | | Blur | | | | Weather | | | | Digital | | | |
|---|---|---|---|---|---|---|---|---|---|---|---|---|---|---|---|---|
| | | Gauss | Shot | Imp | Defoc | Glass | Mot | Zoom | Frost | Snow | Fog | Bright | Cont | Elas | Pix | JPEG |
| *Supervised* | 90 | 32.9 | 30.5 | 28.6 | 35.3 | 25.3 | 36.2 | 36.2 | 34.9 | 30.1 | 42.9 | 65.0 | 35.7 | 42.9 | 45.6 | 53.0 |
| SimCLR | 1000 | 29.1 | 26.3 | 17.3 | 22.1 | 14.7 | 20.0 | 18.6 | 27.2 | 33.3 | 46.2 | 59.7 | 53.9 | 31.0 | 24.2 | 43.9 |
| MoCo (He et al., 2020) | 200 | 29.9 | 26.5 | 10.2 | 26.1 | 24.3 | 33.0 | 20.7 | 32.4 | 25.2 | 28.1 | 52.2 | 47.0 | 43.3 | 35.8 | 40.3 |
| MoCo-v2 (Chen et al., 2020c) | 200 | 51.8 | 50.2 | 36.3 | 48.2 | 44.1 | 50.4 | 36.1 | 50.2 | 40.4 | 44.8 | 63.7 | 58.1 | 58.1 | 58.1 | 52.9 |
| MoCo-v2 (Chen et al., 2020c) | 800 | 56.2 | 54.8 | 39.9 | 52.6 | 48.7 | 58.1 | 40.1 | 53.9 | 45.6 | 51.4 | 67.1 | 62.1 | 61.9 | 61.7 | 56.7 |
| SynCo (ours) | 200 | 52.3 | 50.9 | 34.8 | 48.8 | 44.7 | 51.3 | 36.5 | 49.7 | 39.9 | 44.0 | 63.7 | 58.3 | 58.3 | 58.3 | 52.8 |
| SynCo (ours) | 800 | 57.5 | 56.3 | 40.9 | 53.1 | 49.7 | 57.3 | 41.6 | 53.6 | 44.9 | 49.8 | 66.8 | 62.0 | 61.9 | 61.0 | 55.8 |

and OOD benchmarks, while surpassing SimCLR on OOD datasets such as ImageNet-Sketch (19.2% top-1 accuracy) and ImageNet-R (28.7% top-1 accuracy). Table 12 further highlights SynCo's strong performance across all corruption categories in ImageNet-C, including noise, blur, weather, and digital corruptions. SynCo consistently outperforms MoCo across these categories, demonstrating its ability to maintain high accuracy under a wide range of corruptions. At 800 epochs, SynCo achieves similar performance to MoCo-v2.

### C.4 Adversarial Robustness

We evaluate the adversarial robustness of SynCo by testing against a comprehensive suite of adversarial attacks. Following standard practices in adversarial machine learning (Madry et al., 2018), we assess model performance against both white-box and black-box attacks on the ImageNet validation set. All attacks are implemented using the torchattacks library (Kim, 2020)[7], with evaluations conducted using a ResNet-50 backbone.

Our evaluation includes gradient-based attacks: Fast Gradient Sign Method (FGSM) (Goodfellow et al., 2014) with $\varepsilon = 8/255$, and Projected Gradient Descent (PGD) (Madry et al., 2018) with $\varepsilon = 8/255$, $\alpha = 2/255$, and 10 steps. We also evaluate against optimization-based attacks: Carlini & Wagner (C&W) (Carlini & Wagner, 2017) with confidence $\kappa = 0$, 50 optimization steps, learning rate of 0.01, and initial constant $c = 10^{-4}$. Additionally, we test black-box attacks, including score-based and decision-based methods: Square Attack (Andriushchenko et al., 2020) with $\ell_\infty$ norm and 1,000 queries, and Auto Attack (Croce & Hein, 2020) using $\ell_\infty$ norm. Furthermore, we include advanced perturbation methods: Translation-Invariant FGSM (TIFGSM) (Dong et al., 2019) with $\varepsilon = 8/255$, $\alpha = 2/255$, and 10 steps, and One-Pixel Attack (Su et al., 2019) limited to single-pixel modifications with 10 steps.

Table 13: Top-1 accuracy (in %) under various adversarial attacks on ImageNet validation set using ResNet-50. We reproduce MoCo and MoCo-v2 linear probing since no checkpoints are available.

| Method | Epochs | Clean | FGSM | PGD | C&W | Square | Auto | TIFGSM | OnePixel |
|---|---|---|---|---|---|---|---|---|---|
| *Supervised* | 90 | 76.15 | 23.47 | 0.28 | 16.19 | 11.52 | 0.21 | 4.37 | 73.79 |
| MoCo (He et al., 2020) | 200 | 60.86 | 15.75 | 0.08 | 9.40 | 9.98 | 0.05 | 7.33 | 57.40 |
| MoCo-v2 (Chen et al., 2020c) | 200 | 67.77 | 23.75 | 0.32 | 17.08 | 13.94 | 0.23 | 5.36 | 65.16 |
| MoCo-v2 (Chen et al., 2020c) | 800 | 71.06 | 30.79 | 0.53 | 22.99 | 18.89 | 0.34 | 8.29 | 68.69 |
| SynCo (ours) | 200 | 68.13 | 24.70 | 0.33 | 17.87 | 14.73 | 0.24 | 6.24 | 65.71 |
| SynCo (ours) | 800 | 70.72 | 31.67 | 0.48 | 22.90 | 18.66 | 0.34 | 8.00 | 68.45 |

Results in Table 13 highlight SynCo's strong adversarial robustness across a diverse set of attacks. At 200 epochs, SynCo outperforms MoCo and MoCo-v2 on clean accuracy (68.13%) and demonstrates higher resilience against FGSM (24.70%) and PGD (0.33%) attacks, reflecting its ability to withstand gradient-based perturbations better. Furthermore, SynCo achieves comparable results to MoCo-v2 on optimization-based attacks like C&W (17.87%) and Square Attack (14.73%), while surpassing MoCo in all categories. At

---

[7]Available at: https://github.com/Harry24k/adversarial-attacks-pytorch.

800 epochs, SynCo continues to exhibit competitive performance, achieving parity with MoCo-v2 on clean accuracy (70.72% vs. 71.06%) and similar or slightly better robustness to most attacks.

### C.5   Class Average t-SNE Visualization

We examine the distribution of ImageNet concepts in SynCo's feature space. For each ImageNet class, we compute the average feature vector from its validation images. We apply t-SNE (van der Maaten & Hinton, 2008) with a perplexity of 30 and learning rate of 200 for 1000 iterations. Figure 22 and Figure 23 reveal that SynCo learns meaningful semantic structures: similar animal species naturally cluster together, e.g., spider, barn spider, garden spider, tarantula, wolf spider, and black widow cluster together (bottom right), while digital clock, digital watch, and dial telephone form another coherent group (right mid). The visualization at 800 epochs (Figure 23) shows coherent clusters as well, where e.g., Yorkshire terrier, silky terrier, and Australian terrier cluster together (right mid). We also perform the same analysis for MoCo (He et al., 2020) with 200 epochs of pretraining (Figure 24) and MoCo-v2 (Chen et al., 2020c) with both 200 epochs (Figure 25) and 800 epochs (Figure 26) of pretraining for comparison. Additionally, we include the results from a supervised model trained on ImageNet for comparison (Figure 27).

### C.6   GradCAM Visualization

To gain deeper insights into the regions SynCo focuses on during feature extraction, we utilize GradCAM (Selvaraju et al., 2019) to visualize the model's attention. Attention maps are generated from the final residual block of the ResNet-50 backbone. Figure 10 presents GradCAM visualizations for various ImageNet validation images, comparing SynCo pretrained for 200 epochs and 800 epochs alongside supervised models. The heatmaps reveal that SynCo effectively attends to discriminative object parts and regions, demonstrating its ability to learn meaningful semantic features without supervision.

### C.7   UMAP Visualization

To better understand the feature representations learned by SynCo, we perform Uniform Manifold Approximation and Projection (UMAP) (McInnes et al., 2020) on feature embeddings extracted from the validation set. UMAP reduces high-dimensional data to two dimensions, allowing for a qualitative evaluation of class separability. We considered three configurations based on the number of classes: the first 40, the first 100, and all 1000 classes from ImageNet. Figures 11 and 12 illustrate the results for models pretrained for 200 and 800 epochs, respectively. We also present UMAP visualizations for MoCo with 200 epochs of pretraining (Figure 13) and MoCo-v2 with both 200 epochs (Figure 14) and 800 epochs (Figure 15) of pretraining for comparison. For comparison, we also include UMAP visualizations of features from a supervised model trained on ImageNet (Figure 16).

### C.8   Nearest Neighbor Retrieval

To analyze the semantic consistency of SynCo's learned representations, we perform nearest neighbor retrieval using the following process. We extract 2048-dimensional feature vectors from both ImageNet training and validation sets using the pretrained ResNet-50 backbone with the classification layer removed, applying average pooling to the final convolutional outputs. Using these embeddings, we randomly select query images from the validation set and find their nearest neighbors from the training set memory bank using cosine distance. Since the nearest neighbor is typically the same image in the memory bank, we analyze neighbors #2 through #6. Results shown in Figure 17 demonstrate how SynCo effectively clusters semantically similar images after 200 and 800 epochs of pretraining. We observe that the retrieved neighbors share similar semantic concepts, textures and object poses with the query image.

## D   Ablation Studies

In this section, we perform ablation studies of SynCo on ImageNet-100 and CIFAR-100. For ImageNet we use a ResNet-50, while for CIFAR-100 we use a modified ResNet-18. We compare our method with the baseline

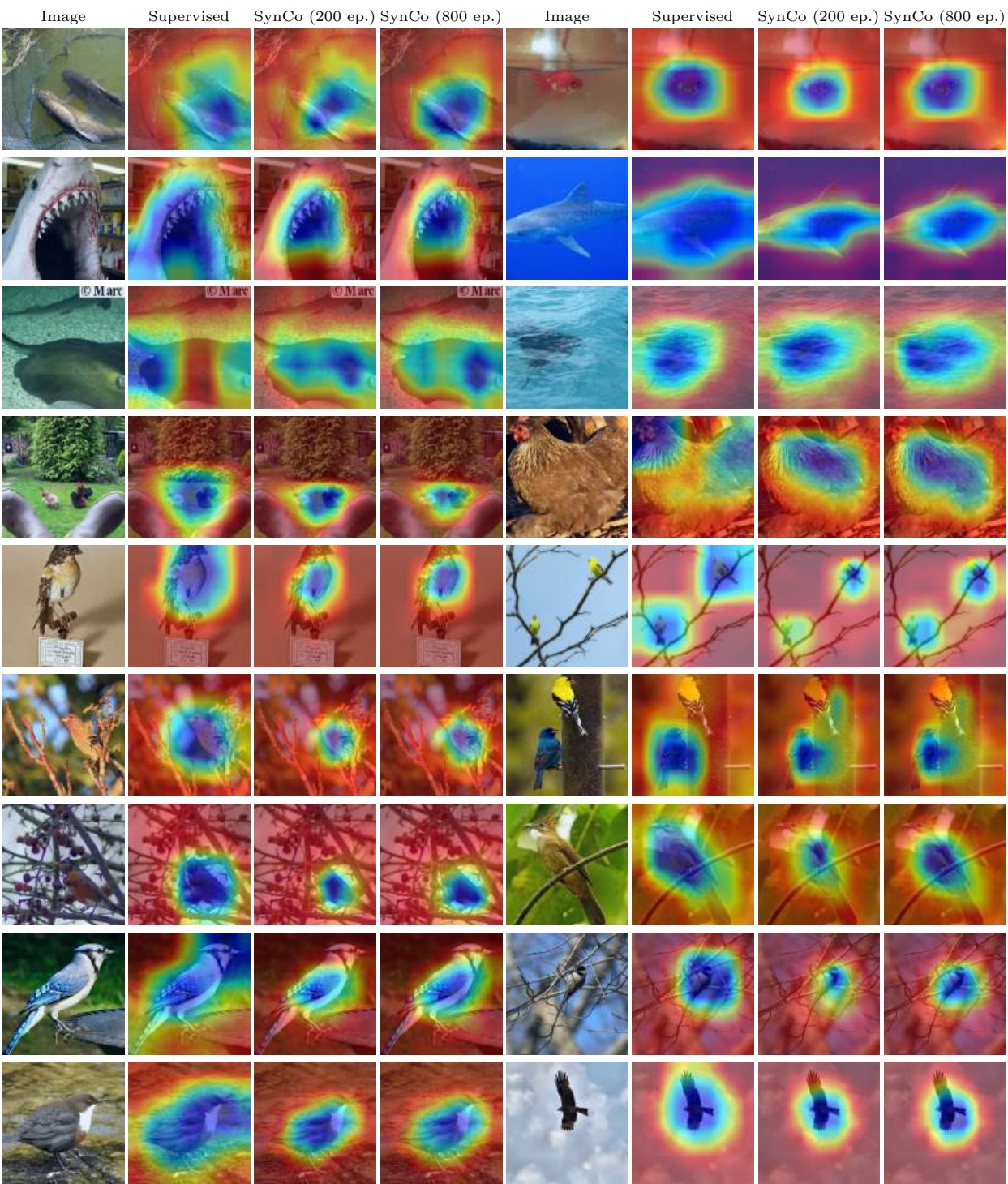

Figure 10: GradCAM visualizations of ImageNet validation set comparing different models: supervised training and SynCo pretrained for 200 and 800 epochs.

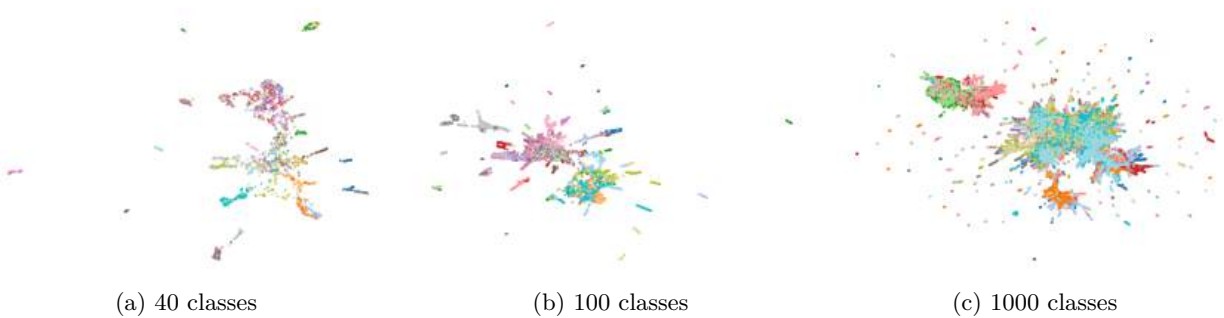

(a) 40 classes      (b) 100 classes      (c) 1000 classes

Figure 11: UMAP visualizations of features extracted from SynCo pretrained for 200 epochs. The visualizations correspond to 40, 100, and 1000 classes of ImageNet validation set.

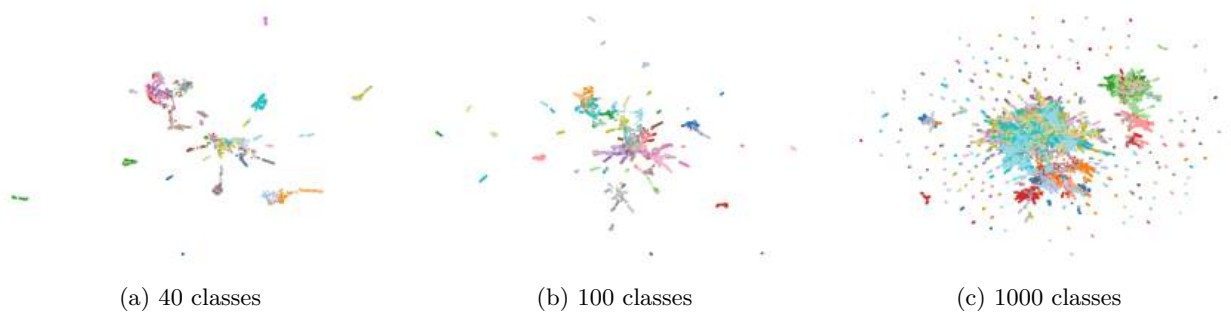

(a) 40 classes      (b) 100 classes      (c) 1000 classes

Figure 12: UMAP visualizations of features extracted from SynCo pretrained for 800 epochs. The visualizations correspond to 40, 100, and 1000 classes of ImageNet validation set.

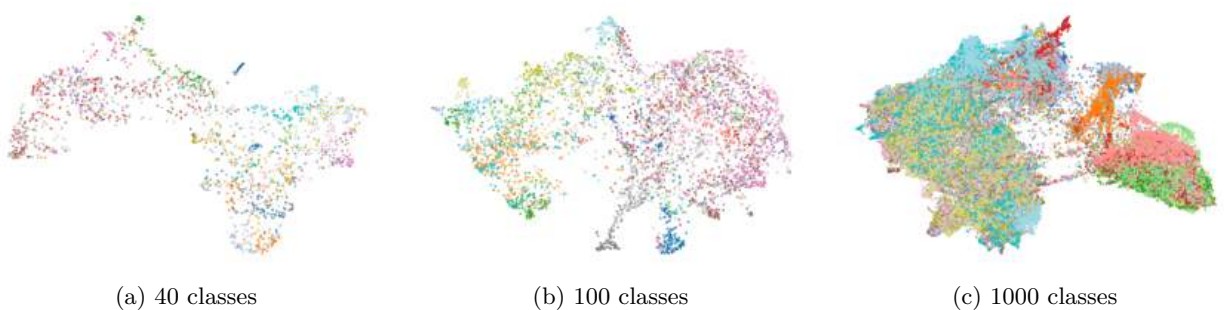

(a) 40 classes      (b) 100 classes      (c) 1000 classes

Figure 13: UMAP visualizations of features extracted from MoCo (He et al., 2020) pretrained for 200 epochs. The visualizations correspond to 40, 100, and 1000 classes of ImageNet validation set.

of MoCo-v2, showing how SynCo improves performance through synthetic hard negatives. Our experiments analyze the impact of different negative types, hyperparameter sensitivity, and queue size variations. We did not search for the optimal combination of negative types, instead opting to ablate each type individually and all together, as even without considering hyperparameters, testing all possible combinations of the 6 negative types would require evaluating 63 different configurations ($2^6 - 1$), which would be computationally prohibitive.

### D.1 Ablation Study on ImageNet-100

First, we perform ablations studies on ImageNet-100 for 100-way classification. Specifically, we ablate SynCo's hyperparameters $\sigma, \delta, \eta$, types (1 to 6), and the effect of queue size $K$ to pretraining. The results of our ablations are presented in Tables 14 to 16. Our findings consistently demonstrate that various SynCo configurations outperform the MoCo-v2 baseline.

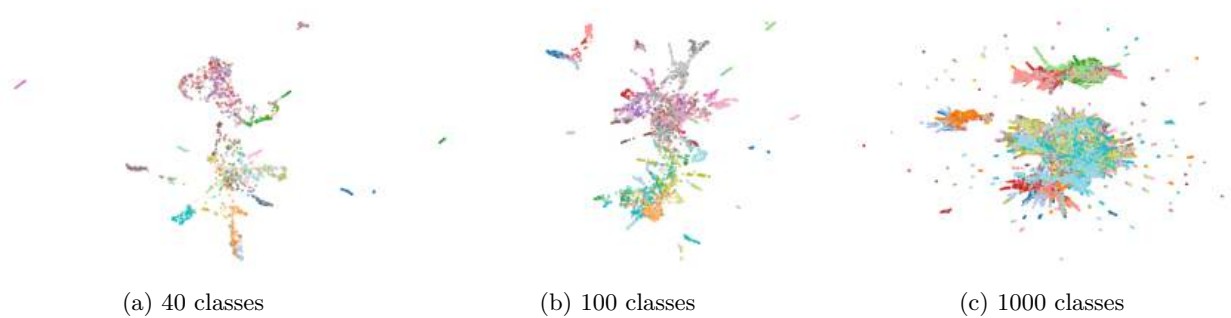

(a) 40 classes        (b) 100 classes        (c) 1000 classes

Figure 14: UMAP visualizations of features extracted from MoCo-v2 (Chen et al., 2020c) pretrained for 200 epochs. The visualizations correspond to 40, 100, and 1000 classes of ImageNet validation set.

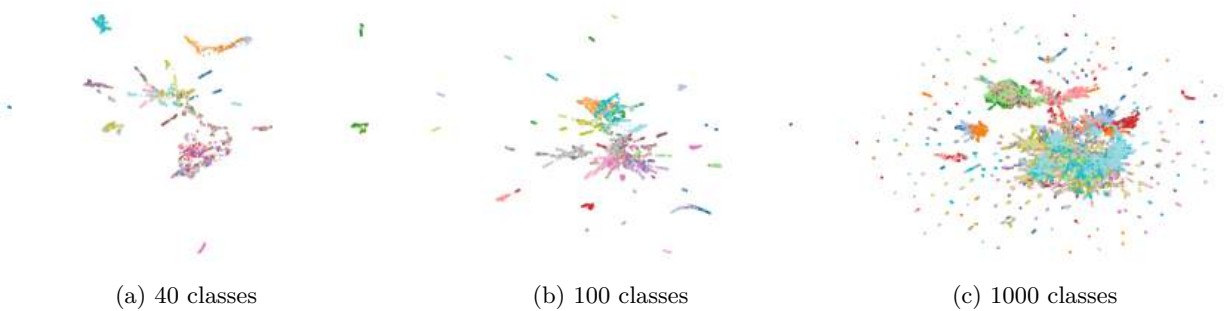

(a) 40 classes        (b) 100 classes        (c) 1000 classes

Figure 15: UMAP visualizations of features extracted from MoCo-v2 (Chen et al., 2020c) pretrained for 800 epochs. The visualizations correspond to 40, 100, and 1000 classes of ImageNet validation set.

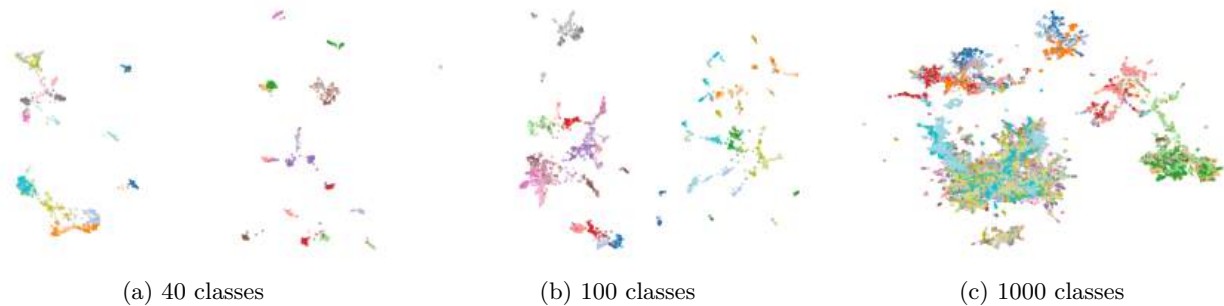

(a) 40 classes        (b) 100 classes        (c) 1000 classes

Figure 16: UMAP visualizations of features extracted from the supervised model. The plots show feature distributions for 40, 100, and 1000 classes from the ImageNet validation set.

**Ablation on hyperparameters.** We conducted ablations on the parameters $\sigma$, $\delta$, and $\eta$ of SynCo's type 4, type 5, and type 6 negatives, respectively. The results, presented in Table 14, show that varying these parameters does not lead to significant differences in performance. This suggests that SynCo is robust across a wide range of values for $\sigma$, $\delta$, $\eta$.

**Ablation on types.** We evaluate the impact of each synthetic hard negative type on pretraining. For this, we select the top $N = 1024$ hardest negatives and generate $N_i = 256$, $i = 1, 2, \ldots, 6$ negatives. We train SynCo without hard negatives (which is equivalent to MoCo-v2) for 100 epochs and measure top-1 and top-5 accuracy. Subsequently, we train using each type of hard negative individually, and then using all six types in combination (which is equivelant to SynCo). The results of these ablations are presented in Table 15. We see that every SynCo configuration outperform the MoCo-v2 baseline.

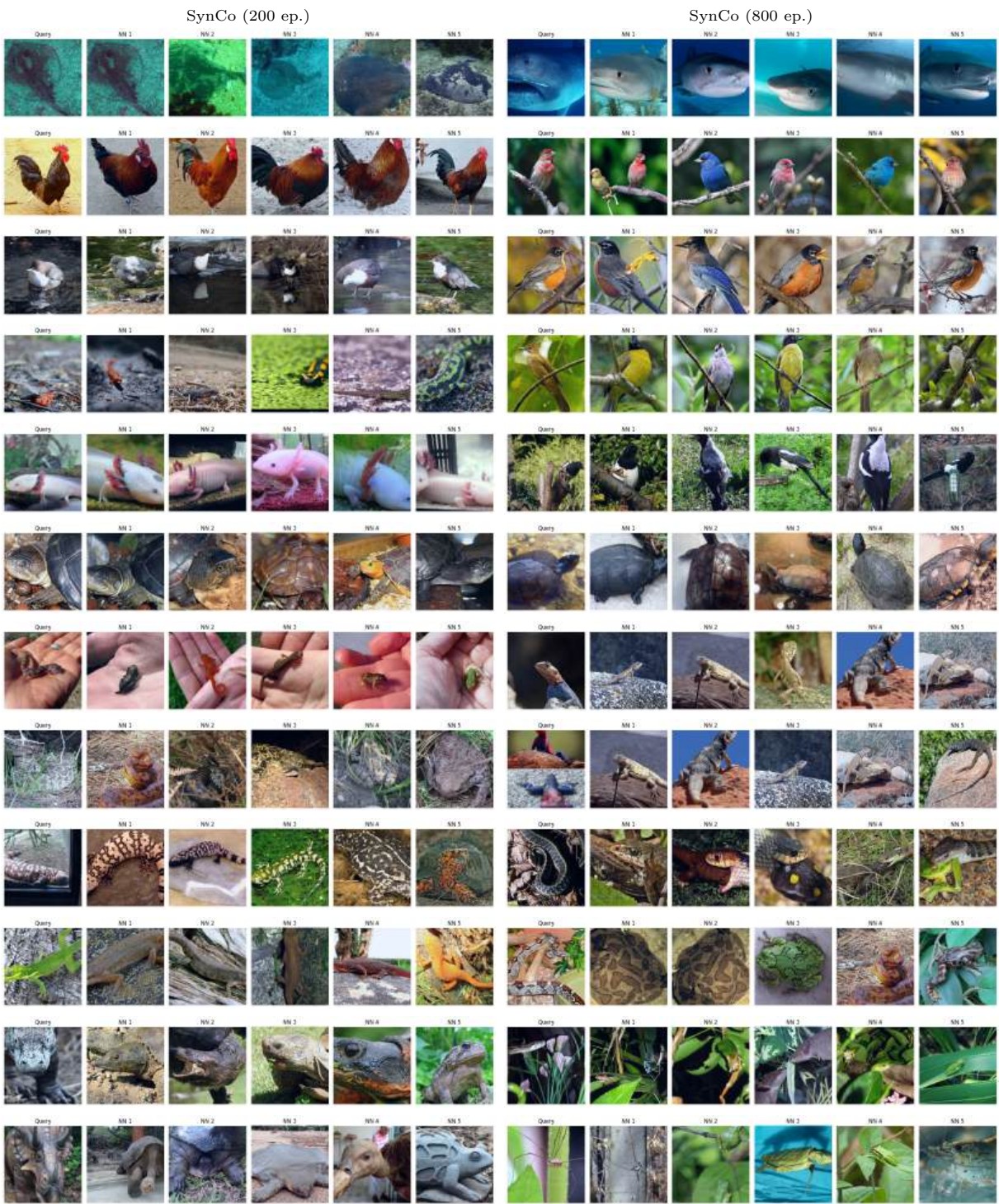

Figure 17: Visualization of nearest neighbors in the embedding space for SynCo pretrained at 200 epochs (left) and 800 epochs (right). Each row corresponds to a query image and its top-5 nearest neighbors in the respective embedding spaces.

Table 14: Top-1 and top-5 accuracies (in %) under linear evaluation on ImageNet-100 with 100 epochs of pretraining using ResNet-50. We ablate different values of SynCo's hyperparameters $\sigma$, $\delta$, and $\eta$.

|  | Value | Top-1 | Top-5 |
|---|---|---|---|
| | 0.01 | 48.20 | 74.26 |
| $\sigma$ | 0.05 | 48.36 | 73.84 |
| | 0.10 | 47.62 | 73.50 |
| | 0.01 | 48.12 | 74.46 |
| $\delta$ | 0.05 | 48.88 | 74.72 |
| | 0.10 | 48.04 | 73.72 |
| | 0.01 | 48.06 | 74.00 |
| $\eta$ | 0.05 | 47.16 | 74.14 |
| | 0.10 | 47.76 | 74.06 |

Table 15: Top-1 and top-5 accuracies (in %) under linear evaluation on ImageNet-100 with 100 epochs of pretraining using ResNet-50. We evaluate different types of hard negatives standalone (Type 1-6), MoCo-v2, and SynCo that employs all types combined.

| Method | Top-1 | Top-5 |
|---|---|---|
| MoCo-v2 (Chen et al., 2020c) | 47.74 | 73.90 |
| Type 1 | 48.22 | 73.92 |
| Type 2 | 48.46 | 74.10 |
| Type 3 | 48.22 | 73.86 |
| Type 4 | 48.20 | 74.26 |
| Type 5 | 48.12 | 74.46 |
| Type 6 | 48.06 | 74.00 |
| SynCo (ours) | **48.42** | **74.16** |

**Ablation on queue size.** We investigate the effect of queue size $\mathcal{Q}$ on performance. We train SynCo and MoCo-v2 with reduced queue sizes. Our results, presented in Table 16, reveal that SynCo performs comparably to MoCo-v2 across various queue sizes. With smaller queues, SynCo underperforms compared to MoCo-v2. This can be attributed to the fact that the total generated synthetic negatives are too hard for the task and harm performance, a finding that is also observed in (Kalantidis et al., 2020). However, as the queue increases, SynCo performs on par with MoCo-v2. At the largest queue size tested, SynCo outperforms MoCo-v2.

Table 16: Top-1 accuracies (in %) under linear evaluation on ImageNet-100 with 100 epochs of pretraining using ResNet-50. We ablate different queue sizes $K$ comparing MoCo-v2 and SynCo.

| Method | Queue size $K$ | | | | | |
|---|---|---|---|---|---|---|
| | 4k | 8k | 16k | 32k | 65k | 131k |
| MoCo-v2 (Chen et al., 2020c) | 50.10 | 50.50 | 49.32 | 48.02 | 47.74 | 47.60 |
| SynCo (ours) | 48.30 | 48.50 | 49.40 | 48.08 | 48.42 | 48.50 |

## D.2 Ablation Study on CIFAR-100

Secondly, we perform additional ablation studies on CIFAR-100 (Krizhevsky, 2009) for $32 \times 32$ images for 100-way classification, chosen for its computational efficiency while maintaining sufficient complexity for meaningful ablations. We ablate the same paramters as in Appendix D.1, with the addition of $N_i, N$, and batch size. We use the same settings as previously discussed with the following differences. We adopt a ResNet-18 (512 output units) (He et al., 2015) architecture without the final classification layer, replacing the original $7 \times 7$ convolutional layer (`conv1`) with a $3 \times 3$ convolution that has a stride of 1 and removing the initial max pooling layer (`maxpool`). The batch size for CIFAR-100 is set to 256, using a single NVIDIA RTX

6000 GPU, and the total training duration is set to 1,000 epochs. Unless stated otherwise, we use $K = 16k$. We report both top-1 and top-5 accuracies as percentages on the test set. When training a linear classifier on top of frozen features, we use a learning rate of 3.0.

**Ablation on parameters.** We evaluate the impact of the parameters $\sigma$, $\delta$, and $\eta$ on SynCo's performance, specifically focusing on type 4, type 5, and type 6 negatives. To determine the optimal settings, we empirically test three sets of values for each parameter: $0.1, 0.05, 0.01$. The results, illustrated in Figure 18, indicate that training SynCo with different values of these parameters yields similar performance across all configurations.

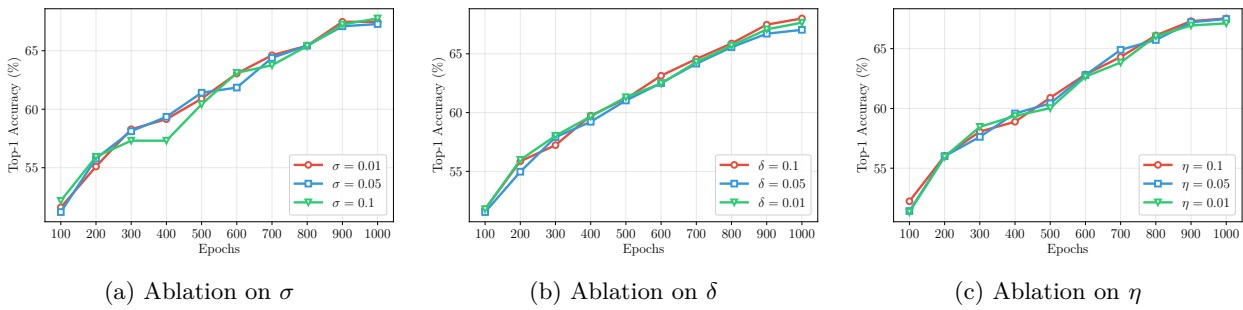

(a) Ablation on $\sigma$        (b) Ablation on $\delta$        (c) Ablation on $\eta$

Figure 18: Top-1 accuracy on CIFAR-100, evaluated every 100 epochs over 1000 epochs of training with varying parameter values. (a) Performance with different $\sigma$ values. (b) Performance with different $\delta$ values. (c) Performance with different $\eta$ values.

**Ablation on types.** We evaluate SynCo by first training without hard negatives (equivalent to MoCo-v2) and then by incorporating each type of hard negative individually, as well as in combination. Additionally, we test different configurations of the number of hard negatives ($N_1$ through $N_6$) to find the optimal settings. The results in Figure 19 show that any incorporation of hard negatives accelerates convergence and improves top-1 accuracy, regardless of type. Increasing the total number of hard negatives beyond $N = 1024$ (e.g., to $N = 2048$) does not further enhance performance, consistent with findings in MoCHI.

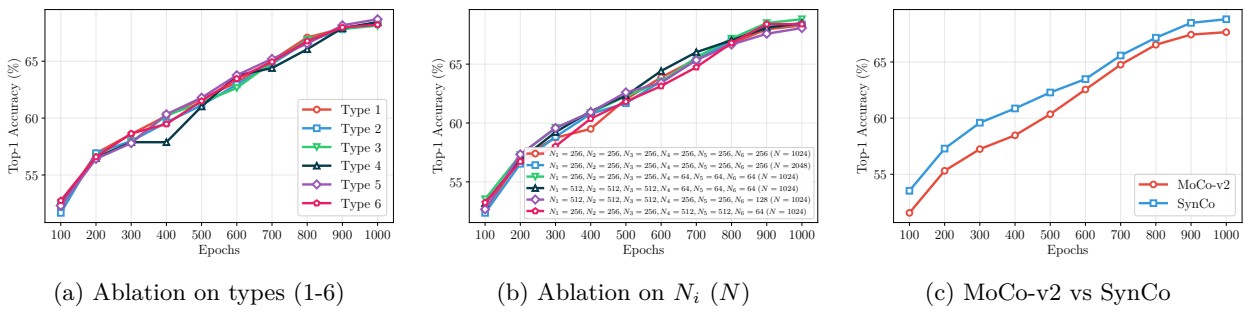

(a) Ablation on types (1-6)      (b) Ablation on $N_i$ ($N$)      (c) MoCo-v2 vs SynCo

Figure 19: Top-1 accuracy on CIFAR-100, evaluated every 100 epochs over 1000 epochs of training. (a) Performance of SynCo with one type of hard negative at a time. (b) Performance of SynCo with varying numbers of hard negatives $N_1$ through $N_6$. Numbers in parentheses represent the maximum $N$ chosen from the queue $\hat{\mathcal{Q}}$. (c) Comparison of SynCo without hard negatives (equivalent to MoCo-v2) and with all hard negatives combined.

**Ablation on queue size.** We evaluate the performance of SynCo across various queue sizes. The results, shown in Figure 20, compare the top-1 accuracy of SynCo and MoCo-v2 across these different queue sizes. SynCo initially performs on par with MoCo-v2, with a minimal performance gap, suggesting that excessively challenging negatives may initially hinder learning efficacy. As the queue size increases, both SynCo and MoCo-v2 show comparable performance, converging further as the queue size maxes out.

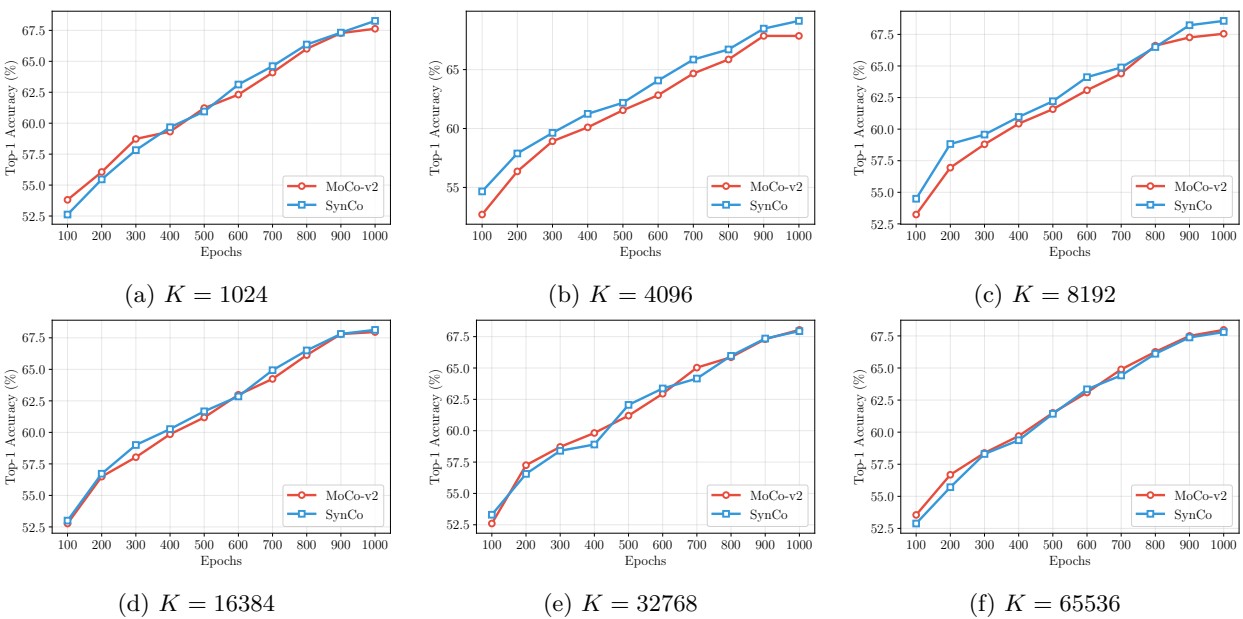

Figure 20: Top-1 accuracy on CIFAR-100, evaluated every 100 epochs over 1000 epochs of training, comparing SynCo and MoCo-v2. (a) With queue size $K = 1024$. (b) With queue size $K = 4096$. (c) With queue size $K = 8192$. (d) With queue size $K = 16384$. (e) With queue size $K = 32768$. (f) With queue size $K = 65536$.

**Ablation on batch size.** We evaluate the effect of varying batch sizes on the performance of SynCo. We tested batch sizes of $64, 128, 256, 512, 1024$, and $4096$. The results are shown in Figure 21. SynCo consistently outperforms MoCo-v2 across all batch sizes, even at the smallest batch size of 64. However, larger batch sizes generally lead to degraded performance for both methods, likely due to the dilution of gradient signals when averaging over larger batches.

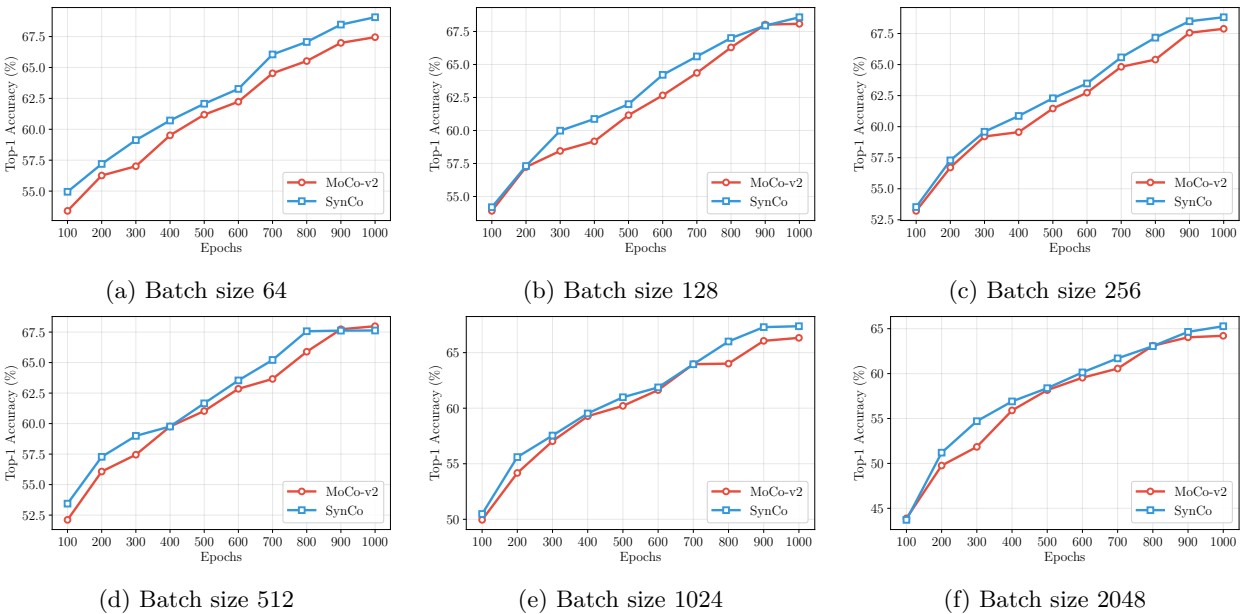

Figure 21: Top-1 accuracy on CIFAR-100, evaluated every 100 epochs over 1000 epochs of training, comparing SynCo with MoCo-v2. (a) With batch size of 64. (b) With batch size of 128. (c) With batch size of 256. (d) With batch size of 512. (e) With batch size of 1024. (f) With batch size of 2048.

# E   Extended Related Work

This section extends our related work discussion by examining two complementary approaches in self-supervised learning, i.e., synthetic feature generation, which enhances model performance with limited labeled data, and generative self-supervised methods, particularly Masked Image Modeling (MIM), which learn by reconstructing or predicting parts of input data.

## E.1   Synthetic Features

Synthetic feature generation is a widely used method to enhance deep learning models, especially with limited labeled data. Adding synthetic features to the representation space improves model generalization and performance. Some methods generate features for unseen classes using generative models (Hariharan & Girshick, 2017; Xian et al., 2018; Schonfeld et al., 2019), while others integrate these into self-supervised and contrastive learning frameworks (Li et al., 2021b; Zhang et al., 2021). This approach has shown success in zero-shot learning (Han et al., 2021). In contrast, our approach directly generates synthetic hard negatives in contrastive learning, without requiring additional generative models.

## E.2   Generative Self-supervised Learning

While the previously discussed methods are discriminative approaches that learn by comparing and distinguishing between different views or instances, another major branch of self-supervised learning takes a generative approach. Generative methods learn by reconstructing or predicting parts of the input data, with MIM emerging as a particularly successful strategy. iGPT (Chen et al., 2020a) demonstrated early success by treating images as sequences for autoregressive prediction, followed by BEiT (Bao et al., 2022) and BEiT-v2 (Peng et al., 2022) which adapted BERT-style (Devlin et al., 2018) masked prediction to vision. MAE (He et al., 2021) showed that aggressive masking of image patches (up to 75%) creates an effective self-supervised task, while SimMIM (Xie et al., 2022) simplified the approach with a lightweight prediction head. Various improvements followed: MaskFeat (Wei et al., 2023) predicted HOG features instead of pixels, Context Autoencoder (Chen et al., 2023) leveraged contextual information, and MSN (Assran et al., 2022) combined masking with siamese networks. Recent work has focused on efficiency and effectiveness through approaches like SiamMAE (Gupta et al., 2023), MixMAE (Liu et al., 2023a), PixMIM (Liu et al., 2023b), and TinyMIM (Ren et al., 2023). The latest developments include CropMAE (Eymaël et al., 2024) with efficient siamese cropped autoencoders and ColorMAE (Hinojosa et al., 2024) exploring data-independent masking strategies. These generative approaches differ fundamentally from discriminative methods by learning to predict or reconstruct missing information rather than comparing different views or instances, offering a complementary approach to self-supervised visual learning.

# F   Discussion

In this section, we examine several critical aspects of our work: the rationale behind comparing with MoCo-based approaches rather than other self-supervised methods; the underlying mechanisms of each synthetic hard negative type and their contribution to model generalization; SynCo's role in model regularization and optimization strategies; the broader implications for other domains like text and audio; connections to classical self-supervised learning foundations; limitations in our hyperparameter analysis due to computational constraints; potential extensions to stronger frameworks like Vision Transformers and larger architectures; and possible adaptations to SimCLR's in-batch negative sampling approach. Through this comprehensive discussion, we aim to provide deeper insights into SynCo's effectiveness, limitations, and future research directions while contextualizing our contributions within the broader landscape of self-supervised learning.

**On the fairness of comparisons.**   Methods such as BYOL (Grill et al., 2020), BT (Zbontar et al., 2021), SwAV (Caron et al., 2020), and VICRegL (Bardes et al., 2022a) incorporate additional *tricks*, including larger projection heads, multi-crop augmentation, and longer training, which significantly improve their performance. However, these improvements stem from architectural and training modifications rather than solely from their core learning mechanisms. In contrast, our approach focuses on demonstrating the effectiveness of synthetic

negative generation within a simpler framework, without relying on such tricks. Therefore, a fair comparison should be made against MoCo-based approaches (MoCo-v2 (Chen et al., 2020c), MoCHI (Kalantidis et al., 2020), PCL (Li et al., 2021a), DCL (Yeh et al., 2022)), which share similar architectural choices and training procedures, ensuring an equitable evaluation of our contributions.

**Intuition of synthetic hard negatives.** Each SynCo strategy improves model generalization through challenging contrasts. Type 1 interpolates between query and hard negatives, increasing sample diversity throughout training. Type 2 extrapolates beyond the query, pushing representation space boundaries and improving robustness to difficult contrasts. Type 3 combines pairs of hard negatives, encouraging more generalized and robust feature learning. Type 4 injects Gaussian noise, promoting invariance to minor feature fluctuations and enhancing generalization. Type 5 modifies embeddings based on similarity gradients, refining discriminatory power by directing the model towards harder negatives. Type 6 applies adversarial perturbations, creating the most challenging contrasts to distinguish deceptively similar samples.

**Using hard negatives for model regularization.** SynCo addresses existing limitations by generating hard negatives on-the-fly, reducing computational overhead while maintaining diverse contrasts. It regularizes the network through synthetic hard negatives, aligning with vicinal risk minimization (Chapelle et al., 2000). This encourages learning robust features over memorization, addressing poor generalization common in empirical risk minimization (Vapnik, 1998; Zhang et al., 2017a). The diverse synthetic negatives create a comprehensive learning environment, improving generalization across datasets and tasks. This approach reduces overfitting and enhances robustness to data variations, leading to more robust representations (Zhang et al., 2017a).

**Considerations for parameter tuning and optimal use of synthetic negatives.** In our experiments, we searched for optimal parameters for each type of synthetic negative. We used all types of synthetic negatives to demonstrate overall improvements. However, incorporating fewer synthetic negatives, rather than all, could potentially lead to higher accuracy. Here, our focus was on proposing the concept of synthetic negatives rather than searching for the optimal combination. The optimal combination of synthetic negatives depends on the specific dataset and task. We also did not exhaustively search for the most effective number of synthetic negatives to generate. Instead, we conducted initial experiments to assess their effectiveness.

**Implications of synthetic hard negatives in broader contexts.** The introduction of synthetic hard negatives in contrastive learning not only improves model performance in traditional image classification and detection tasks but also holds potential for applications beyond the current scope. Synthetic hard negatives can be adapted for various modalities, including text, audio, and multi-modal learning environments. For instance, in natural language processing, generating challenging negative samples could enhance tasks such as sentence similarity, text classification, and language translation. Similarly, in audio processing, synthetic hard negatives might improve tasks like speaker recognition or audio event detection. Moreover, the adaptability of synthetic hard negatives opens up possibilities for future research into domain adaptation and transfer learning. By incorporating domain-specific hard negatives, models can better generalize across different domains, addressing the challenge of domain shift in practical applications. This adaptability also suggests that synthetic hard negatives could be a crucial component in developing more robust, generalizable machine learning systems across various fields.

**Bridging classical and modern self-supervised learning.** While many of the foundational works in self-supervised learning date back several years, the principles and challenges they address remain fundamentally relevant. Our work demonstrates how these established frameworks can be improved through synthetic hard negative generation. This approach bridges classical self-supervised learning techniques with contemporary needs for more efficient and robust representation learning. The method's success in improving performance across various tasks suggests that self-supervised learning, particularly when augmented with synthetic samples in the embedding space, continues to offer valuable directions for advancing AI systems. As the field moves toward more efficient and generalizable learning approaches, techniques that can work with limited labeled data while maintaining strong performance become increasingly important.

**Limitations on hyperparameter analysis.** While our experiments demonstrate SynCo's effectiveness across various configurations, our comprehensive hyperparameter analysis is primarily based on CIFAR-100, with findings extended to ImageNet. Due to computational constraints, we cannot exhaustively ablate these parameters on larger datasets. Nevertheless, our results show that SynCo is remarkably robust to variations in hyperparameters ($\sigma$, $\delta$, $\eta$) and the number of synthetic negatives ($N_i$, $N$). This versatility suggests that even without dataset-specific optimization, SynCo can achieve strong performance with default parameters. Future work could explore more fine-grained parameter tuning for specific datasets and domains.

**Potential extensions using stronger frameworks.** While our current implementation is built on MoCo-v2 (Chen et al., 2020c) for computational efficiency (requiring only 4 GPUs), SynCo's principles could be integrated with more advanced frameworks. Using larger projection and prediction (Grill et al., 2020), incorporating multi-crop augmentation (Caron et al., 2020), or leveraging advanced architectures (Xie et al., 2021b) could potentially boost performance further. The method could also be extended to Vision Transformers (Xie et al., 2021b; Chen et al., 2021). However, these advanced frameworks typically require significant computational resources ($> 8$ GPUs), making them currently impractical for our experimental validation. Future work could explore these extensions when more computational resources are available.

**Potential extension to SimCLR.** While our method is built upon MoCo-v2 (Chen et al., 2020c)'s memory bank, the concept of synthetic hard negatives could be adapted to SimCLR (Chen et al., 2020b)'s in-batch negative sampling approach. Instead of generating synthetic negatives from memory bank features, one could generate them from in-batch features. However, SimCLR typically requires very large batch sizes (4096) and significant GPU resources (8+ GPUs) to achieve competitive performance, making such an implementation computationally prohibitive for our current experimental validation. This remains an interesting direction for future research.

## G   Checkpoint Availability

The pre-trained model checkpoints for models trained on the ImageNet ILSVRC-2012 dataset are available for download: 200-epoch model (top-1 linear evaluation accuracy 68.1%) and 800-epoch model (top-1 linear evaluation accuracy 70.7%).

## H   Broader Impact

The presented research should be categorized as research in the field of unsupervised learning. This work may inspire new algorithms, theoretical, and experimental investigation. The algorithm presented here can be used for many different vision applications and a particular use may have both positive or negative impacts, which is known as the dual use problem. Besides, as vision datasets could be biased, the representation learned by SynCo could be susceptible to replicate these biases.

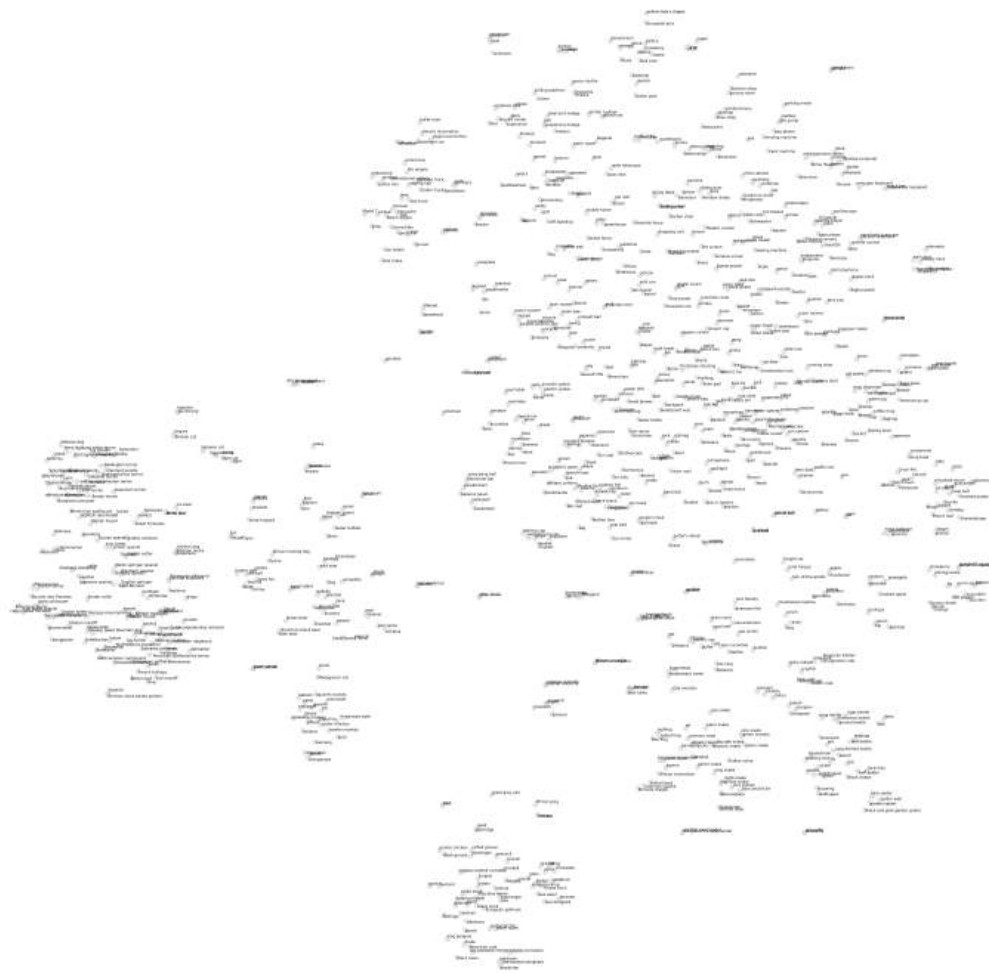

Figure 22: t-SNE visualization of ImageNet class embeddings in SynCo's feature space. Each point represents the average feature vector of validation set images for one class. The visualization reveals semantic clustering, with similar concepts appearing close together. SynCo is pretrained for 200 epochs.

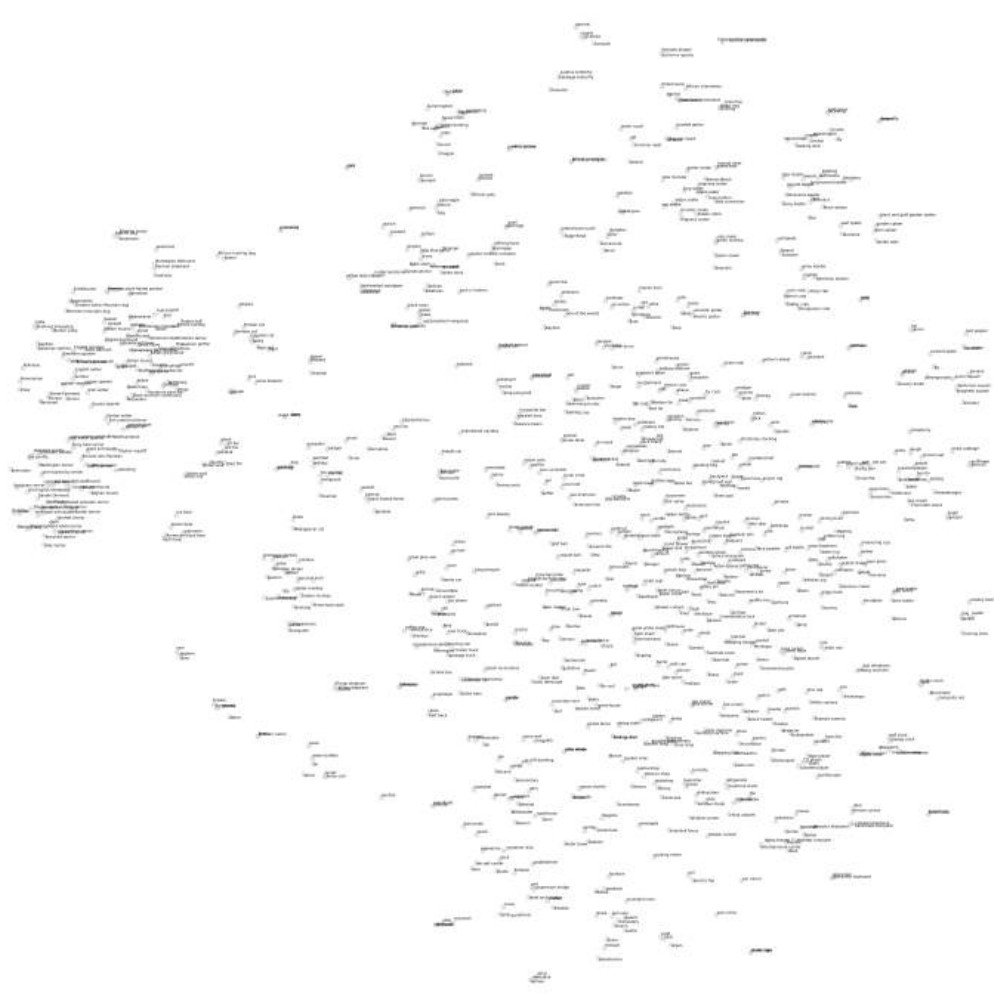

Figure 23: t-SNE visualization of ImageNet class embeddings in SynCo's feature space after 800 epochs of pretraining. Each point represents the average feature vector of validation set images for one class.

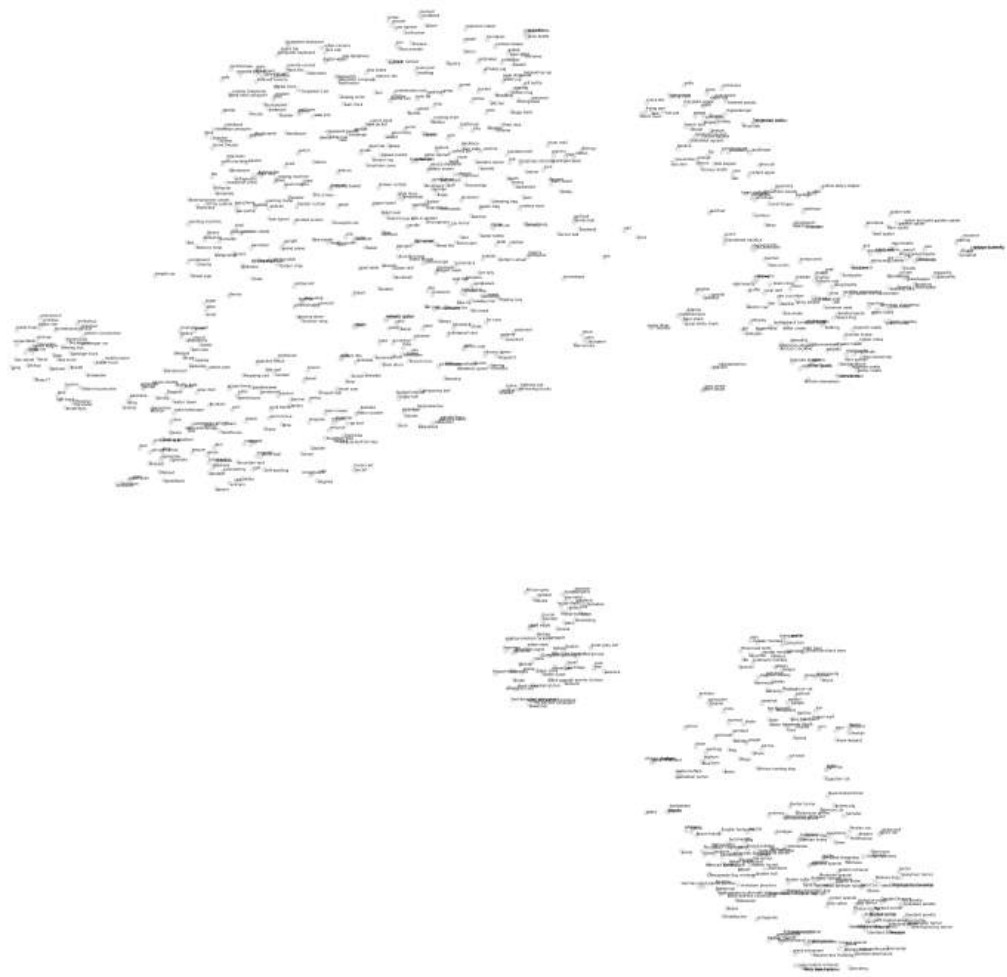

Figure 24: t-SNE visualization of ImageNet class embeddings in MoCo's (He et al., 2020) feature space after 200 epochs of pretraining. Each point represents the average feature vector of validation set images for one class.

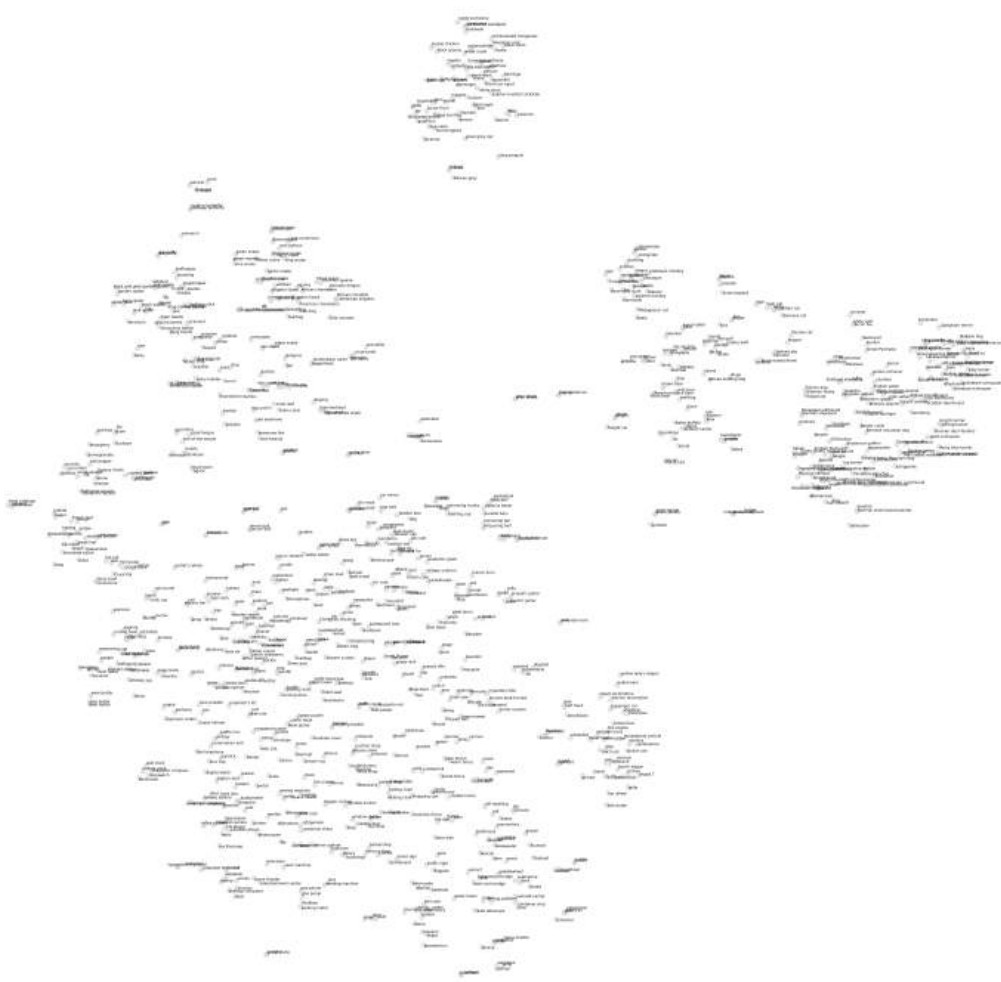

Figure 25: t-SNE visualization of ImageNet class embeddings in MoCo-v2's (Chen et al., 2020c) feature space after 200 epochs of pretraining. Each point represents the average feature vector of validation set images for one class.

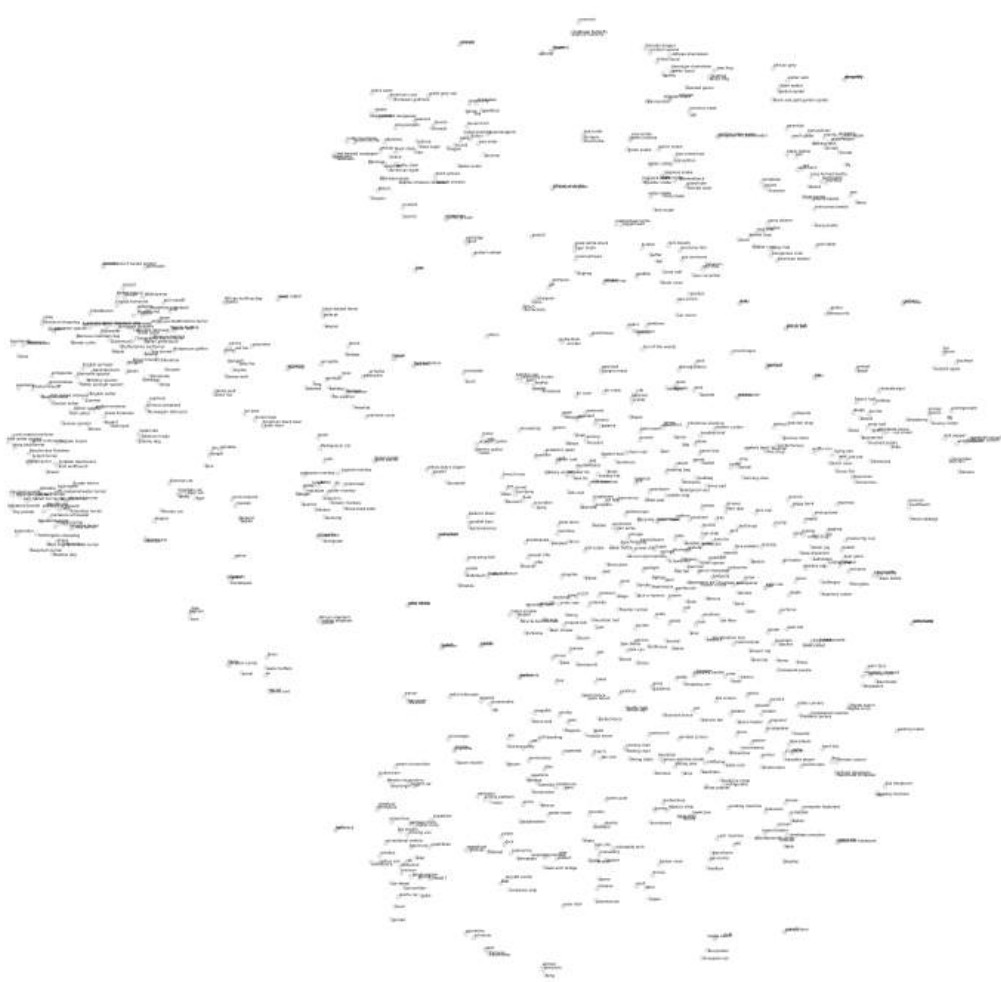

Figure 26: t-SNE visualization of ImageNet class embeddings in MoCo-v2's (Chen et al., 2020c) feature space after 800 epochs of pretraining. Each point represents the average feature vector of validation set images for one class.

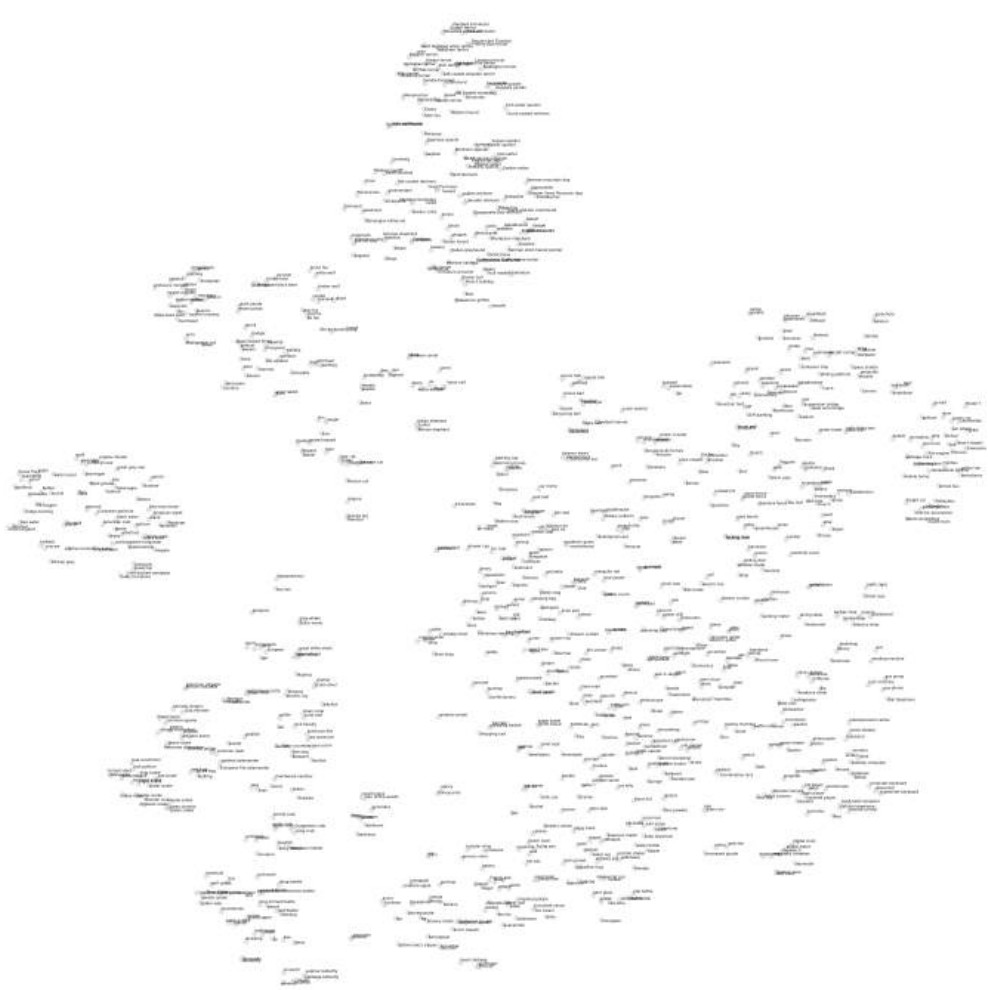

Figure 27: t-SNE visualization of ImageNet class embeddings in the supervised feature space. Each point corresponds to the mean feature vector of validation images belonging to a single class.

