# OpenReview forum: "SynCo: Synthetic Hard Negatives for Contrastive Visual Representation Learning"
_TMLR — Rejected by TMLR_

### Review · Reviewer_mSJr · 2025-07-15

**Summary Of Contributions:**

The paper explores six different ways of mining negatives to regularize contrastive learning methods (in particular MoCo v2).  The paper presents extensive results that evaluate the proposed mining mechanisms in linear evaluations in ImageNet and detection tasks on PASCAL VOC and COCO.

**Audience:**

Yes

**Broader Impact Concerns:**

No concerns

**Claims And Evidence:**

Yes

**Requested Changes:**

- Update Fig. 4 with different colors or patterns for the synthetic negatives since it is hard to distinguish the shades of blue alone.
- What the authors mean by "appropriate tasks" should be clearly defined (e.g., P6 in the type 2 description).
- I understand that space may be limited.  However, I recommend the authors to consider moving some of the results for the different negative mining techniques ablations into the main paper.  It is natural to wonder how they compare to each other instead of just seeing the final result.

**Strengths And Weaknesses:**

Strengths:
- The idea is simple and easy to follow which aids on its reproduction.  The six augmentations are straightforward, while the descriptions could be improved and streamlined in the paper, with some effort from the reader it is possible to understand what each augmentation is doing.
- The experiments are extensive and show improvements over the compared methods.
- There are ablations over the different mining techniques (although on the appendix) as well as other hyperparameters for the method.

Weaknesses:
- The experiments rely on ResNet backbones, while the existing methods have moved into using ViT-based backbones (e.g., DINO and iBOT)
- What an "appropriate task" is is not clearly defined in the paper.

---

### Review · Reviewer_L9Lm · 2025-07-21

**Summary Of Contributions:**

The paper introduces SynCo, a self-supervised visual contrastive learning framework that integrates six distinct hard-negative generation techniques. Authors conduct experiment results show SynCo’s effectiveness over some existing contrastive learning methods.

**Audience:**

No

**Claims And Evidence:**

No

**Requested Changes:**

The authors should answer the questions and refine the manuscript addressing the weaknesses mentioned above.

**Strengths And Weaknesses:**

### Strength
1. The paper proposes SynCo, a visual contrastive learning framework that integrates six distinct hard-negative generation techniques, offering a methodical and unified strategy.
2. Experiment results show SynCo’s effectiveness over some existing contrastive learning methods.

### Weakness
1. The main experimental results only benchmark against methods from 2022 or earlier. To strengthen the paper’s proposed ideas, the authors should include more recent methods (from the last 2–3 years) in the main result tables.
2. Several proposed techniques echo many prior work (e.g., MoCHI [1], AdCo [2] and many of their following works). The authors should provide deeper insight, both intuitively and experimentally, with direct comparisons against these baselines.
3. Techniques 1, 2, and 3 appear closely related, as do 4 and 6. This overlap increases hyperparameter complexity and computational cost. While Table 15 offers a breakdown of individual technique effectiveness, the authors should include ablation studies on combined subsets (e.g., {1,4}, {2,5}, etc.) and discuss whether all six are needed together.
4. The full framework introduces a high-dimensional tuning space. The authors should present ablations and provide practical guidelines on hyperparameter selection, trade-offs, and sensitivity when multiple generators are used simultaneously.
5. Figures such as Figure 2 and Figures 22–27 are low resolution and hard to read. These should be redrawn or replaced with higher-resolution versions to improve clarity and readability.

[1] Yannis Kalantidis, Mert Bulent Sariyildiz, Noe Pion, Philippe Weinzaepfel, and Diane Larlus. Hard negative mixing for contrastive learning, 2020.

[2] Qianjiang Hu, Xiao Wang, Wei Hu, and Guo-Jun Qi. 2021. Adco: Adversarial contrast for efficient learning of unsupervised representations from self-trained negative adversaries. In Proceedings of the IEEE/CVF Conference on Computer Vision and Pattern Recognition. 1074–1083.

---

### Review · Reviewer_sc5X · 2025-07-22

**Summary Of Contributions:**

This paper improved MoCo-v2 by introducing synthetic hard negatives to make the representation learning be more challenging so that the learned representation is more discriminative for the downstream tasks, such as image classification and object detection. This work introduced 6 types of negative examples. The experimental results show consistent improvement over original MoCoV2 on image classification and object detection.

**Audience:**

Yes

**Claims And Evidence:**

Yes

**Requested Changes:**

1. Please address the weaknesses.

2. In Table 15, using only Type 2 achieves similar performance to the one using all, how does this justify the benefits of other types of negative samples?

3. As shown in experimental results, non- MoCo-based method, like SwAV achieved significant better performance than SynCo, how could the proposed method be used on all other methods?

4. At table 4 and 5, why using the model trained with 200 epochs for detection task instead of 800 epochs one?

5. it is still unclear to me that why stop the hard examples at epoch 400 will achieve better results, if so, how to the users to determine when to stop using those hard examples?

**Strengths And Weaknesses:**

Strengths
1. In constrastive learning, crafting the proxy task is the key of representation learning, by introducing a harder task usually could provide better learned representation.

2. The paper is well-written and easy to understand and follow.

Weaknesses
1. The survey on hard negatives is weak, there are many literatures that aims to craft hard negatives are not included and those should be used to compared to proposed method. [1][2][3][4]

References:
- [1] CONTRASTIVE LEARNING WITH
HARD NEGATIVE SAMPLES, ICLR 2021
- [2] Hard-Negative Sampling for Contrastive Learning: Optimal Representation Geometry and Neural- vs Dimensional-Collapse, TMLR 2025
- [3] Does Negative Sampling Matter? a Review With Insights Into its Theory and Applications, TPAMI 2024
- [4] Robust Contrastive Learning Using Negative Samples with Diminished Semantics, NeurIPS 2021

2. The improvement over MoCoV2 is marginal for imagenet1k; moreover, without properly tuning SynCo, it is even worse than MoCoV2 (training with 800 epochs).

---

### Decision · Action_Editor_PDFK · 2025-09-05

**Recommendation:** Reject

**Additional Comments:**

None.

**Audience:**

Yes

**Audience Explanation:**

Contrastive learning is an important paradigm in machine learning and modern deep models, and the use of hard negative samples has proven effective in enhancing its performance. A submission addressing this topic is likely to be of significant interest to many researchers in this journal’s readership.

**Claims And Evidence:**

No

**Claims Explanation:**

This paper addresses an important task and presents a well-written study that proposes a unified strategy for handling hard negatives. The experimental results are extensive and demonstrate the efficacy of the proposed method over several existing approaches, making the contribution valuable.

However, the reviewers identified several concerns. The survey of related work on hard negatives is incomplete, with many relevant methods not included or compared against, and the paper would benefit from deeper insights and comparisons with prior studies. While the proposed approach shows improvements, they are marginal in some cases, and the work needs to better justify the benefits of each type of negative sample and explain more clearly how the method interacts with other techniques. Reviewers also raised questions about how users determine when to stop using hard examples and noted the need for practical guidelines on hyperparameter selection. Additional concerns include the limited scope of comparisons, as the experimental study only considered methods from 2022 or earlier, and the need for stronger ablation studies, particularly on combined subsets. Furthermore, the lack of experiments with ViT-based backbones was initially a notable omission.

The rebuttal addressed some of these issues by clarifying how users could stop using hard examples, discussing hyperparameter sensitivity and selection, and providing experiments with ViT-based backbones. While these responses were helpful, they did not fully resolve the main concerns.

The final recommendations from reviewers were two leaning reject and one accept. The key concerns that remain are: (1) the similarity between the proposed hard negative generation techniques and existing methods, (2) the potential difficulty of tuning the multiple hyperparameters introduced, and (3) the unclear interaction of the proposed method with various existing techniques.

Based on the reviews and rebuttal, the AE recommends a decision of rejection. While the paper has notable strengths in presentation, task significance, and experimental effort, the claims are not well-supported because the qualitative and quantitative comparisons are inadequate. That is, this work should address the omission of many prior studies on hard negative crafting, compare them against the proposed method, and provide stronger empirical discussion of other contrastive learning approaches to justify the benefits of the proposed one. Therefore, the need to clarify the relationship to prior studies, strengthen comparative analysis, and address practical applicability outweigh the contributions in its current form. The authors are encouraged to further strengthen the work by addressing the review comments in greater depth.

**Resubmission Of Major Revision:**

The authors may consider submitting a major revision at a later time.